# Rapid, DNA-induced interface swapping by DNA gyrase

**Thomas RM Germe[1]\*[†], Natassja G Bush[1][‡], Victoria M Baskerville[1], Dominik Saman[2], Justin LP Benesch[2], Anthony Maxwell[1,3]**

[1]Department Biochemistry & Metabolism, John Innes Centre, Norwich Research Park, Norwich, United Kingdom; [2]Department of Chemistry, Biochemistry Building, University of Oxford, Oxford, United Kingdom; [3]Department of Molecular Microbiology, John Innes Centre, Norwich Research Park, Norwich, United Kingdom

**\*For correspondence:**
thomas.germe@cuanschutz.edu

**Present address:** [†]Dept. of Biochemistry and Molecular Genetics, University of Colorado Anschutz, Aurora, United States; [‡]Inspiralis Ltd., Innovation Centre, Norwich Research Park, Norwich, United Kingdom

**Competing interest:** The authors declare that no competing interests exist.

**Abstract** DNA gyrase, a ubiquitous bacterial enzyme, is a type IIA topoisomerase formed by heterotetramerisation of 2 GyrA subunits and 2 GyrB subunits, to form the active complex. DNA gyrase can loop DNA around the C-terminal domains (CTDs) of GyrA and pass one DNA duplex through a transient double-strand break (DSB) established in another duplex. This results in the conversion from a positive (+1) to a negative (–1) supercoil, thereby introducing negative super-coiling into the bacterial genome by steps of 2, an activity essential for DNA replication and transcription. The strong protein interface in the GyrA dimer must be broken to allow passage of the transported DNA segment and it is generally assumed that the interface is usually stable and only opens when DNA is transported, to prevent the introduction of deleterious DSBs in the genome. In this paper, we show that DNA gyrase can exchange its DNA-cleaving interfaces between two active heterotetramers. This so-called interface 'swapping' (IS) can occur within a few minutes in solution. We also show that bending of DNA by gyrase is essential for cleavage but not for DNA binding per se and favors IS. Interface swapping is also favored by DNA wrapping and an excess of GyrB. We suggest that proximity, promoted by GyrB oligomerization and binding and wrapping along a length of DNA, between two heterotetramers favors rapid interface swapping. This swapping does not require ATP, occurs in the presence of fluoroquinolones, and raises the possibility of non-homologous recombination solely through gyrase activity. The ability of gyrase to undergo interface swapping explains how gyrase heterodimers, containing a single active-site tyrosine, can carry out double-strand passage reactions and therefore suggests an alternative explanation to the recently proposed 'swivelling' mechanism for DNA gyrase (Gubaev et al., 2016).

## eLife assessment

This is an **important** study on DNA gyrase that provides further evidence for its mode of action via a double-stranded DNA break and against a recently-proposed alternative mechanism. The evidence presented is **solid** and is derived from state-of-the-art techniques. The work casts new light on the interactions that occur between gyrase molecules and will be of interest to biochemists and cell biologists.

## Introduction

The genome of bacterial cells is in a negative-superhelical state through the combined activity of various DNA topoisomerases, enzymes that can alter the linking number between the two DNA strands constituting the bacterial genome (*Bush et al., 2015*; *Vos et al., 2011*). The only enzyme that can introduce negative supercoils is a type IIA topoisomerase, DNA gyrase. An active gyrase

complex is constituted by two GyrA subunits, encoded by the *gyrA* gene, and two GyrB subunits, encoded by the *gyrB* gene, thus forming an $A_2B_2$ heterotramer (***Figure 1b***). The $A_2B_2$ complex can bind a segment of DNA (green, ***Figure 1b***) sandwiched between the two TOPRIM domains of the GyrB subunits and the two tower domains of GyrA and sitting on the winged-helix domain (WHD) domain of GyrA. The enzyme is able to establish a transient double-strand break in this DNA segment through transesterification from a 5′-phosphate in the DNA backbone to a catalytic tyrosine (Y122 in *Escherichia coli* gyrase, ***Figure 1d***). The DNA duplex that extends from the cleaved segment can wrap in a positive loop around the GyrA C-terminal domains (CTDs) and the other extremity of the loop can be passed through the various interfaces of the enzyme, including the DNA break, thereby converting the positive wrap into a negative one, and resulting in the introduction of negative super-coils by a step of 2, once the DNA break has been re-sealed (***Brown and Cozzarelli, 1979***; ***Mizuuchi et al., 1980***; ***Figure 1b***). Therefore, gyrase needs to be able to transport a DNA duplex through the whole enzyme, that is three interfaces (***Figure 1a and b***). The first one is constituted by the ATPase domain at the N-terminus of GyrB, whose dimerization is controlled by the binding of two ATPs and results in the trapping of the transported DNA (T) segment that enters through the open interface. This interface is dubbed the N-gate. The transport of DNA is coupled to the binding and hydrolysis of ATP. The second interface the duplex must go through is the DNA gate, which is constituted by the DNA break itself and a protein-protein interface between the two GyrA subunits (WHD domains). The duplex can then exit the enzyme through the C-gate (***Figure 1b***). There must be some coupling between the interfaces opening and closing since opening them all at the same time would cause the enzyme to come apart and the formation of a permanent double-stranded break (DSB). This is never observed at any detectable level with the purified enzyme in vitro.

It has been suggested that the coupling between ATP binding and hydrolysis acts as a safety mechanism by only allowing DNA transport through the DNA gate and C-gate when the ATP interface (N-gate) is closed (***Bates and Maxwell, 2007***). However, gyrase can perform strand passage without ATP (and therefore with the N-gate opened), without introducing DSBs (***Gellert et al., 1977***; ***Sugino et al., 1977***), suggesting that some allosteric coupling prevents the DNA gate and the C-gate from being opened at the same time. Consistent with this, the purified GyrA subunit forms stable dimers in solution, whereas the GyrB subunit can be observed as a monomer (see results) in the absence of ATP (***Costenaro et al., 2007***). While the N-gate is controlled by ATP, it is unclear what controls the opening of the DNA gate and the C-gate. However, clear structural evidence obtained, mainly with eukaryotic topoisomerase II, suggests coupling between motion of the WHD domains and the opening of the C-gate, pointing to a coordination between DNA cleavage status and the opening of the C-gate (***Dong and Berger, 2007***; ***Wendorff et al., 2012***). In the case of gyrase, however, no structure with the C-gate open is available so it is difficult to conclude. It has also been suggested that the transported DNA segment itself controls the opening of the various interfaces for instance by 'pushing open' the DNA-gate (***Bax et al., 2019***; ***Roca, 2004***).

DNA gyrase is the target for the clinically important antibiotics, the fluoroquinolones. These compounds bind the enzyme and stabilize the cleaved state, the so-called 'cleavage complex'. The resulting DSBs are deleterious for the bacterial cell and are thought to be the basis for their antibacterial properties. However, in vitro, the cleavage complexes stabilized by fluoroquinolones are reversible, suggesting that the enzyme subunits do not come apart. It is therefore unclear how the reversible cleavage complexes are converted in vivo to irreversible DNA lesions. However, treatment of bacterial cells by fluoroquinolones clearly results in DSBs (***Bush et al., 2020***; ***Chen et al., 1996***). In eukaryotic cells, pathways exist for the removal of the covalently bound topoisomerase II enzyme and exposure of the DNA lesion to regular DNA repair mechanisms (***Pommier et al., 2014***). In bacterial cells such pathways are not well known (but see ***Huang et al., 2021***).

Some work has also suggested that in *Escherichia coli* gyrase cleavage complexes can induce illegitimate recombination between two DNA segments (***Ikeda, 1994***; ***Ikeda et al., 1981***; ***Ikeda et al., 2004***). One possible mechanism being the exchange of subunits between two cleavage complexes, followed by re-ligation. However, these experiments were conducted in a complex cellular extract and whether the observed recombination occurs through subunit exchange is not known. It has been shown that treatment with fluoroquinolones is associated with genomic rearrangements (***López and Blázquez, 2009***; ***López et al., 2007***), potentially playing a role in their cytotoxicity, but again the exact mechanism remains unclear.

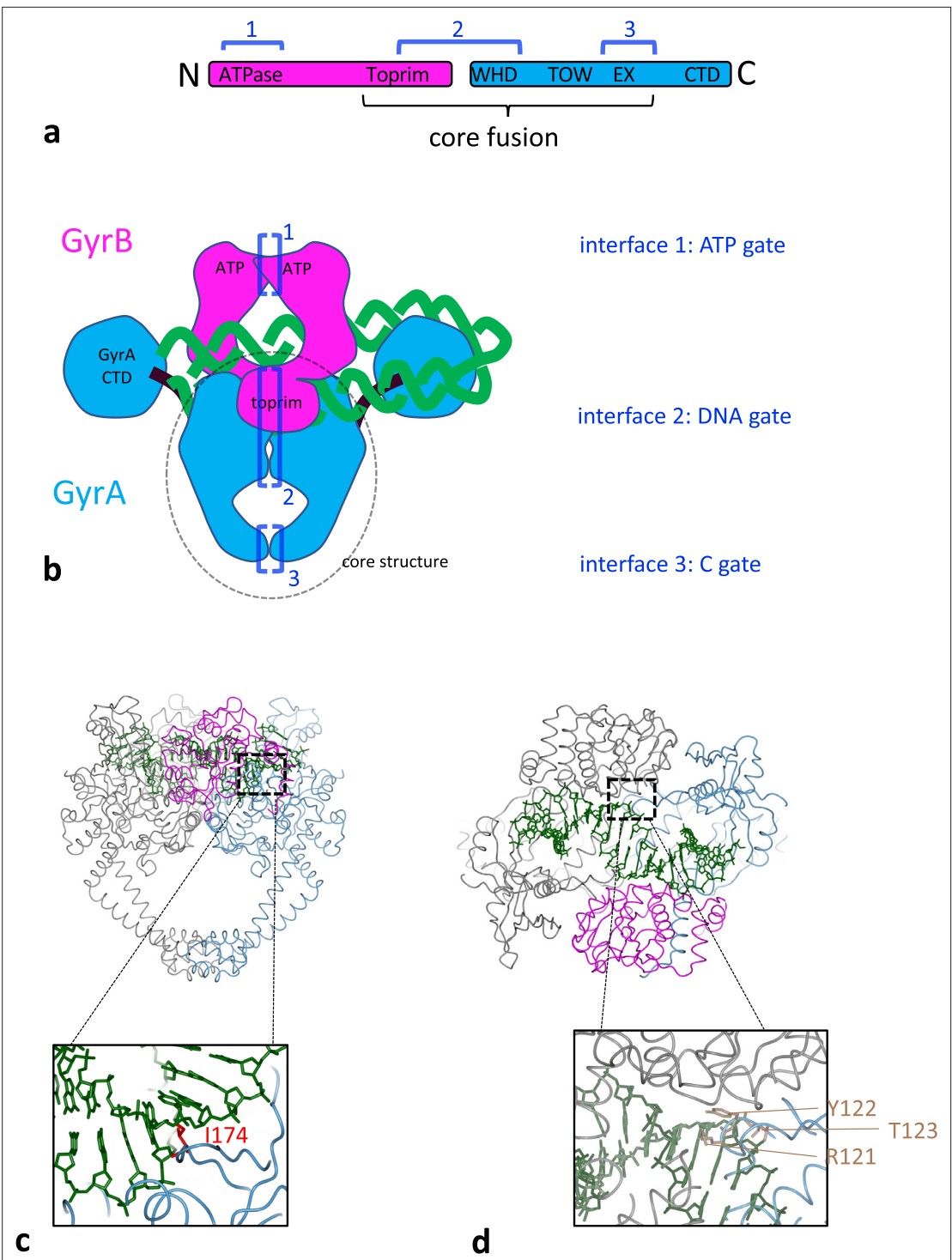

**Figure 1.** Structural organisation of DNA gyrase. (**a**) Schematic of DNA gyrase peptidic sequences. GyrB is in purple and GyrA in blue. The core fusion construct used in structural studies is indicated. The sequences engaging in one of the three interfaces are indicated by dark blue brackets. (**b**) Cartoon rendering of the DNA gyrase heterotetramer, color-coded as above. The part of the enzyme included in the *S. aureus* core fusion structure is circled with a dashed line. Various domains are indicated, and the three interfaces the transported DNA go through are are labeled 1–3 in the order of passage. (**c**) Side view of *S. aureus* DNA gyrase core fusion structure (6FQV) with bound DNA (green) (only the TOPRIM domain here). The C2 symmetry mate is rendered in gray. The inset shows the location of isoleucine 174 in red (*E. coli* numbering) which is inserted between base +8 and+9 (from scissile phosphate) and introduces a kink in the duplex. (**d**) Top view of the same structure. The inset shows in light brown the catalytic tyrosine sandwiched between an arginine and a threonine and forming the triad RYT.

*Figure 1 continued on next page*

*Figure 1 continued*

The online version of this article includes the following figure supplement(s) for figure 1:

**Figure supplement 1.** Gel filtration and SDS-PAGE analysis of our heterodimer preparation.

**Figure supplement 2.** Schematic of possible sequential interface swapping mechanism leading to complete subunit exchange.

In this work we purify *E. coli* gyrase complexes with a heterodimeric GyrA interface as pioneered in the Klostermeier group (*Gubaev et al., 2016*). The complexes are purified as a dimer of GyrA and a fusion of GyrB and GyrA (BA fusion), encoded by different plasmids and thus allowing the introduction of targeted mutations on one side only of the complex (*Figure 1—figure supplement 1*). An active enzyme is reconstituted by the addition of free purified GyrB. We show that mutating the catalytic tyrosine on one side completely abolishes cleavage on this side of the complex and severely compromises negative supercoiling activity by removing the ability to double-strand cleave DNA. We show that the complex can nevertheless reconstitute double-strand cleavage activity, underlying the reconstitution of negative supercoiling activity by strand-passage through the double-strand break. We suggest that this reconstitution of double-strand cleavage activity occurs when the DNA gate interface of a gyrase dimer is open, and then re-forms by association between one subunit from one dimer with another subunit coming from another dimer (*Figure 1—figure supplement 2*). The exchanged interface functionality would be conserved as the contacts and configuration of the exchanged interface remains identical to the unexchanged interface, and therefore this process, which we dub 'interface swapping' (IS), can reconstitute double-strand cleavage when the exchanged interface presents 2 active tyrosine. Interface swapping can lead to whole 'subunit exchange' depending on the interfaces involved (*Figure 1b*, *Figure 1—figure supplement 2*). For instance, swapping interfaces 2 and 3 leads to the whole GyrA subunit to be exchanged, whereas to exchange a BA fusion subunit all three interfaces needs to be exchanged. In addition, we show that interface swapping is favored by the presence of the free GyrB subunit and that these free GyrB subunits can multimerize and pull the heterodimers into higher order multimers. We show that mutating isoleucine 174 in GyrA (*E. coli* numbering, *Figure 1c*) abolishes DNA cleavage but not DNA binding and only lowers wrapping activity. Introducing the I174G mutation on one side of the heterodimer severely reduces cleavage activity suggesting that proper DNA bending and/or wrapping favors interface swapping. Altogether, our data suggest that DNA binding and wrapping and the accumulation of gyrase in higher order oligomers can favor rapid interface exchange between two active gyrase complexes brought into proximity. We discuss the possible mechanism and suggest that gyrase can recombine two DNA duplexes through interface exchange.

## Results

### Preparation of stable GyrA-GyrBA heterodimers

Our initial aim was to study the cooperativity between the two sides of the dyad axis. Inspired by the Klostermeier group (*Gubaev et al., 2016*), we developed a procedure to express and purify heterodimers of GyrA and a GyrB-GyrA fusion construct. The fusion is N-terminally tagged with 6xHis. Co-expressing both constructs in *E. coli* yields a reasonable amount of heterodimers (around 1 mg for 1 L of culture). They are purified by a combination of $Ni^{2+}$ affinity, ion exchange and size exclusion chromatography (see Materials and Methods). The quality of the preparations was assessed by analytical gel filtration (*Figure 1—figure supplement 1c*), Blue-Native PAGE, Mass Photometry and native mass spectrometry (*Figure 2*, *Figure 2—figure supplement 1* and *Figure 2—figure supplement 2a*).

### Gel-filtration analysis of gyrase heterodimers

Our heterodimer preparation showed a single-peak on a gel-filtration column, distinct from the GyrA dimer peak. An SDS-PAGE analysis of the peak showed two bands, whose sizes are consistent with GyrA and the GyrBA fusion construct (*Figure 1—figure supplement 1*). We also compared our different preparations by Blue-native PAGE. GyrA protein migrates mainly as dimers, but tetramers (slower migrating band marked 'A₄ tetramer', *Figure 2a*, *Figure 2—figure supplement 2a*) and potentially higher order oligomers are also observed. GyrB migrates as a monomer but a ladder of bands with slower migration is also observed (*Figure 2a* and *Figure 2—figure supplement 2a*), showing the

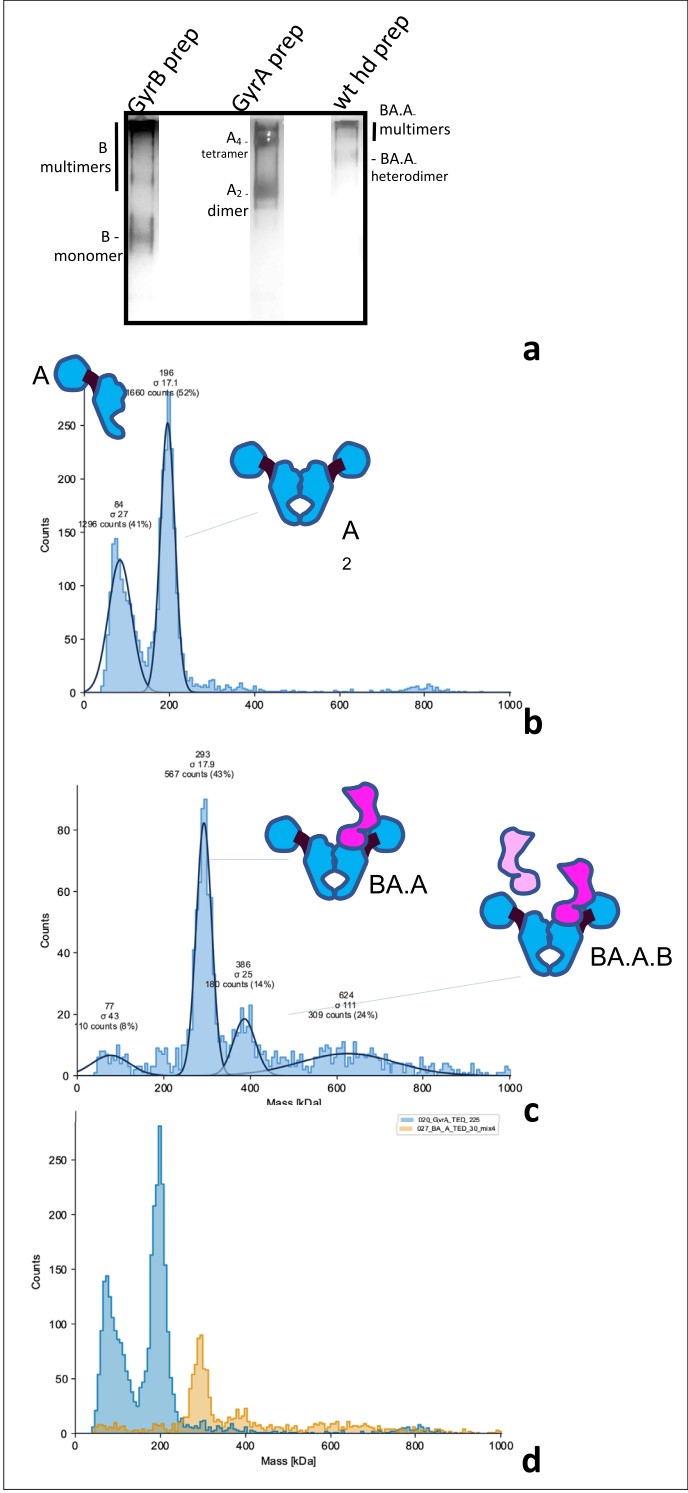

**Figure 2.** Analysis of GyrA, GyrB, and heterodimers preparations. (**a**) Blue-Native PAGE of our GyrB, GyrA and wild-type heterodimer preparation. The bands constituted by GyrB monomers, GyrA dimers and heterodimers are indicated and were visualized by silver staining. The migration pattern was ascertained by comparison to a native marker and is consistent with mass photometry profiles below. (**b**) Mass Photometry profile of the GyrA preparation. The histogram shows the counts of collisions events plotted against the scattering intensity, which is proportional to the MW and is calibrated against urease; the abscissa shows the molecular mass in kDa. The instrument fits the observed peaks to gaussian curves (continuous black lines) and the mean (MW) and deviation (σ) of the fitted curves are indicated on top of the peak, alongside the total count for the peak and the percentage

*Figure 2 continued on next page*

*Figure 2 continued*

of counts assigned to the peak with respect to the total number of fitted events. (**c**) Mass Photometry profile of our wild-type heterodimer preparation, as above. Note that the lower the peak count, the higher the deviation. We detect a main peak at approximately the expected size for a heterodimer. (**d**) Superimposition of the two profiles (GyrA dimer in blue and heterodimer in orange) showing the difference in mass between the heterodimer and the GyrA dimer. The two profiles were collected on the same day, in the same buffer and with the same calibration.

The online version of this article includes the following figure supplement(s) for figure 2:

**Figure supplement 1.** Analysis of GyrA and BA$_F$A preparation by native mass spectrometry.

**Figure supplement 2.** Blue-Native PAGE and western blot of the GyrA/GyrA59 heterodimers.

**Figure supplement 3.** BA.A heterodimer and GyrA dimer mixing experiment; done to assess sensitivity of mass photometry to contaminating GyrA dimers mixed with BA.A heterodimers.

---

propensity of *E. coli* GyrB to multimerize. Compared to these preparations, our heterodimer preparation migrates mainly in a band intermediate between the GyrA dimer and the GyrA tetramer, again consistent with the theoretical MW of the complex. Some higher-order complexes (i.e. multimers) are also detected (slower migrating species) in the heterodimer preparation, consistent with the observation that both GyrA dimers and GyrB monomers can multimerize, presumably conferring upon the heterodimer the ability to multimerize.

## Mass photometry analysis of gyrase heterodimers

We also used mass photometry to analyze our preparations. This technique analyzes the light scattered by molecules in solution when they hit a glass surface (*Young et al., 2018*). The scattering intensity correlates with the molecular weight of the particles. The resulting histograms can be fitted to a gaussian and calibrated with a known preparation (urease here). This allows the estimation of the MW of particles in the preparation. Our GyrA preparation shows two peaks, one at around ~200 kDa, consistent with the GyrA dimer and one at around ~80–90 kDa, consistent with the GyrA monomer (*Figure 2b and d*). This is surprising considering that the monomer is not detected by either gel filtration or Blue-native PAGE (*Figure 2a*, *Figure 1—figure supplement 1c* and *Figure 2—figure supplement 2a*).

For mass photometry, the samples are diluted to an appropriate concentration to allow a limited number of collision events on the glass slide. This dilution varies depending on the sample being analyzed, presumably due to difference in the way different complexes and protein interact with the glass surface. In the case of GyrA the dilution was important, and the concentration of the mass photometry sample is around 25 nM. At such dilution some dissociation is expected, and we calculated a dissociation constant of 5.7 nM. This is close to the affinity seen with antibodies for instance and suggests a strong GyrA interface.

In the case of gel-filtration and blue-native PAGE, the concentration is much higher as the preparation is not diluted (around 400 times higher). Assuming the accuracy of our previously estimated dissociation constant, we calculate that the monomer/dimer ratio should be around 1–100 at this concentration, thus explaining why the monomers are not detected. In the case of our heterodimer preparation, we detect a main peak at the expected MW (286 kDa), very low monomer and no GyrA dimers (*Figure 2c and d*). This shows that our heterodimer preparations are not contaminated with GyrA dimers. Incubating the preparation at 37 °C for 30–60 min did not change the profile, showing that subunit exchange, which would re-form GyrA dimers, does not occur significantly during this period and that the low amount of monomer is not sufficient for solution-based exchange (see below).

The heterodimer preparation is diluted before analysis but not as much as the GyrA preparation, the dilution factor was adjusted for each preparation in order to obtain analyzable data. We cannot calculate a dissociation constant for this sample since GyrA and GyrBA fusion monomer are not detectable. However, assuming that the dissociation constant for the heterodimers is similar to that of the GyrA dimer, we calculate that the monomer/dimer ratio at this concentration should be just below 1 in 10. This appears generally consistent with our mass photometry profile and indicates that the concentration of monomer is very low at working concentrations of DNA gyrase (in the DNA cleavage assay) and a fortiori in vivo. We also detect a peak consistent with a heterotetramer, which could be consistent with either a heterodimer binding a contaminating GyrB monomer (BA.A.B trimer), or more

likely contaminating GyrBA fusion dimers (BA$_2$). The amount remains low and produces low background DNA cleavage in a cleavage assay (see below), which rules out a nuclease contamination. We also detect a complex consistent with a dimer of heterodimers. The heterodimer preparation shows a wider range of sizes, and the maximum number of collisions that can be analyzed is constant. Therefore, the number of collisions assigned to a single species in the sample will be lower than for a more homogeneous sample (like GyrA, see the overlay in *Figure 2d*). This can decrease the sensitivity and increase the error in the molecular weight measurement of minor species. However, we can detect a low amount of BA$_2$ or BA.A.B contamination and therefore any undetected contamination is unlikely to be significant.

To make sure that a potential GyrA contamination can be detected with reasonable sensitivity in the heterodimer preparation, we have also measured preparations of GyrA and BA.A heterodimer mixed at different ratios. We observed that a low amount of GyrA can be detected in the mix (1 in 20), along with the BA.A peak, showing that any contamination with GyrA dimer would be low, below the detection level (*Figure 2—figure supplement 3*). Moreover, the GyrA dimer is detected at a very high sensitivity with mass photometry as evidenced by the high dilution required to achieve the optimal number of collisions on the glass slide. This experiment also shows that mass photometry should not be used to measure absolute relative abundance of different species without careful calibration of mixed samples. Altogether, these data show that our heterodimer preparation is not significantly contaminated by GyrA dimer but has some contamination by either a BA$_2$ homodimer or a BA.A.B trimer. According to our mass photometry data, this contamination accounts for 14% of the preparation (*Figure 2c*), the majority of which is constituted by genuine heterodimers.

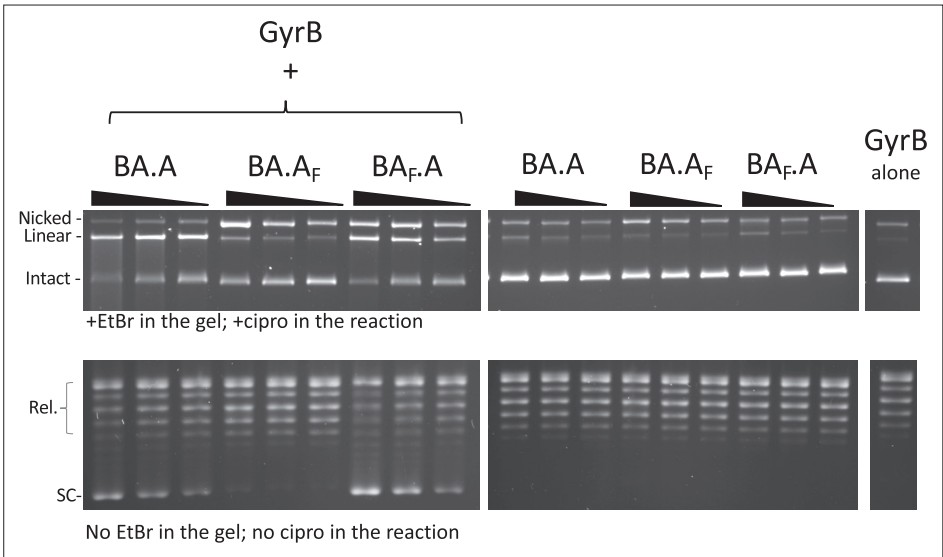

**Figure 3.** Reconstitution of double-strand cleavage activity in the presence of free GyrB. Both cleavage (top) and supercoiling (bottom) assays were performed with BA.A, BA.A$_F$ and BA$_F$.A as indicated. For cleavage assays, 5, 2.5, and 1.25 pmole of heterodimers (three lanes, from left to right for each heterodimer version, the triangle shows increasing dosage of heterodimers) were used. Eight pmole of GyrB were added to reconstitute the activity (top left panel). Omitting GyrB showed only background cleavage (top center panel). GyrB alone (top right panel) showed almost undetectable cleavage activity. For supercoiling assays, the dose of each subunit was reduced, so as to keep the activity limiting in the assay. Four pmole of GyrB was added to 1.25, 0.625, and 0.312 pmole of heterodimer, the triangle showing increasing dose of the heterodimer. Cleavage assays are analyzed with EtBr-containing agarose gels and the cleavage reaction contains 20 µM ciprofloxacin (cipro). Supercoiling assays do not contain cipro in the reaction and are analyzed on agarose gels that do not contain EtBr. The two types of gel/assay are indicated.

The online version of this article includes the following figure supplement(s) for figure 3:

**Figure supplement 1.** BA.A$_F$ can form double-strand cleavage complexes in the presence of GyrB.

**Figure supplement 2.** Cleavage activity of various heterodimer mutants.

### Native mass spectrometry of gyrase heterodimers

We have also analyzed our GyrA and heterodimer preparation by native mass spectrometry (*Figure 2—figure supplement 1*). The GyrA dimer was readily detectable with the expected mass (194 kDa). The profile is entirely consistent with our mass photometry profiles and our BN-PAGE experiment, showing a GyrA monomer (97 kDa), a GyrA tetramer (388 kDa) and even possibly a hexamer. The heterodimer preparation showed a peak at the expected size for a heterodimer (289 kDa) and a GyrA monomer peak (according to the size, 97 kDa). This is consistent with our mass photometry profiles with the exception of the $BA_2$ or BA.A.B contamination, which is not detectable by mass spectrometry.

### Heterodimers with one catalytic tyrosine mutated can reconstitute double-strand cleavage activity

We have tested our heterodimers in both cleavage and supercoiling assays. Free GyrB was added to the heterodimer prep to reconstitute an active enzyme. The wild-type heterodimer displayed robust double-strand cleavage activity in the presence of 20 µM ciprofloxacin and can negatively supercoil DNA (*Figure 3*). We mutated the catalytic tyrosine to a phenylalanine on the GyrA side of the complex ($BA.A_F$) and the resulting enzyme displayed a severely reduced double-strand cleavage activity in the presence of ciprofloxacin and now displayed a robust single-strand cleavage activity. This is reminiscent of results obtained previously (*Gubaev et al., 2016*). This result is consistent with the formation of a cleavage complex with only one catalytic tyrosine. The supercoiling activity was drastically reduced as well, although not completely abolished, again consistent with earlier work (*Gubaev et al., 2016*). In both wild type and mutant, both the cleavage and supercoiling activity are dependent upon the addition of GyrB, as expected. When we introduced the tyrosine mutation on the GyrBA fusion side of the complex ($BA_F.A$) we observed, in addition to the expected single-strand cleavage activity, a very high level of double-strand cleavage, correlating with a restored high supercoiling activity, comparable to the wild type. Both activities were again dependent upon the addition of GyrB.

Cleavage complexes are trapped at the phenol-aqueous interface when performing a phenol extraction of the sample. It is therefore possible to purify nucleic acids that are covalently linked to the enzyme by harvesting nucleic acids trapped at the phenol-aqueous interface (Materials and methods). We observe that the single-strand cleaved DNA produced by the $BA.A_F$ plus GyrB is recovered from the phenol interface, as expected (*Figure 3—figure supplement 1*). We also recovered a significant amount of double-strand cleaved DNA from the phenol interface, suggesting that the ability of this mutant heterodimer to reconstitute double-strand cleavage activity is also present, as in the case of $BA_F.A$ mutant, only much reduced. This double-strand cleavage activity was dependent on the addition of GyrB, which differentiates it from GyrB-independent background double-strand cleavage, due to contaminating GyrBA fusion homodimer. Therefore, both $BA_F.A$ and $BA.A_F$ homodimer mutants can reconstitute double-strand cleavage activity in vitro in the presence of GyrB, albeit at different levels (see discussion for possible reasons for this asymmetry). Consequently, both mutants must be able to reconstitute strand-passage activity, which is consistent with both displaying GyrB-dependent supercoiling activity. The level of supercoiling activity correlates with the level of double-strand cleavage activity, further supporting this interpretation. This double-strand DNA cleavage activity was unexpected and could be interpreted as: (1) an initial contamination of the heterodimer preparation with GyrA dimers, which we have ruled out in the previous sections; (2) an alternative nucleophile from the polypeptidic chain able to catalyze the same transesterification as the catalytic tyrosine, or (3) interface swapping between two heterodimers, reconstituting an active double-strand DNA cleaving interface (*Figure 1b*, *Figure 1—figure supplement 2*).

### The reconstituted double-strand cleavage activity is not due to an alternative nucleophile

We therefore tested the possibility that the reconstituted double-strand cleavage is due to an alternative nucleophile that substitutes for the tyrosine on the mutated side. This nucleophile could be provided by the polypeptidic chain or even by the solvent in the form of a water molecule (like a nuclease). The catalytic tyrosine is surrounded by potential nucleophiles: an arginine on the N side and a threonine on the C side, forming the peptidic sequence RYT. We therefore mutated these residues to leucines and tested them in our heterodimer system. We tested three mutants: the $BA_{RFA}.A$, where the tyrosine is mutated to a phenylalanine and the threonine to an alanine, the $BA_{RLL}.A$, where both

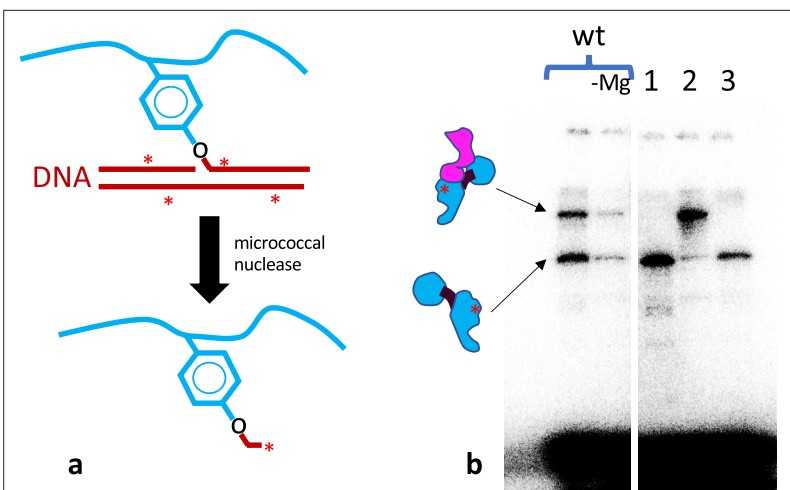

**Figure 4.** Labelling of the nucleophilic amino acid in gyrase. (**a**) Schematic of the radiolabeling of the catalytic tyrosine. The GyrA polypeptidic chain is schematized in blue, with the catalytic tyrosine shown. A cleavage reaction results in a covalent bond between the catalytic tyrosine and a radiolabeled DNA (in red with asterisks). Subsequent digestion with micrococcal nuclease digests most of the DNA and leaves a stub of radiolabeled nucleic acid covalently bound to the catalytic tyrosine. (**b**) The resulting labelled gyrase polypeptide can be analyzed by SDS-PAGE and detected by exposure to a phosphor screen. The cleavage reactions contained 8 pmole of GyrB and 5 pmole of heterodimer. From left to right: wild-type heterodimer with $Mg^{2+}$, without $Mg^{2+}$, $BA_F.A$ (1), $BA.A_F$ (2), and $BA_{LLL}.A$ (3). The upper band is the fusion polypeptide, the lower band is the GyrA polypeptide. On both, the red asterisk shows the approximate position of the radioactive label. The smear at the bottom is the bulk of the digested radiolabeled DNA.

The online version of this article includes the following figure supplement(s) for figure 4:

**Figure supplement 1.** Exonuclease sensitivity of heterodimeric gyrase cleavage products.

tyrosine and threonine are mutated to leucines, and the $BA_{LLL}.A$, where all three are mutated to leucines. All three mutants displayed robust reconstitution of double-strand cleavage activity (**Figure 3—figure supplement 2**). In fact, in all three mutants, the double-strand cleavage was elevated at identical enzyme concentration (**Figure 3—figure supplement 2**, discussed below). Therefore, if an alternative nucleophile was present, it would have to come from somewhere else along the peptidic chain. To completely exclude this possibility, we performed cleavage assays with radiolabeled DNA (**Figure 4**). The subsequent treatment with micrococcal nuclease leaves a stub of radiolabeled nucleic acid covalently linked to the polypeptide that provided the nucleophile. The protein can be then resolved by SDS-PAGE and detected by exposure to a phosphor screen. Only polypeptides with an active nucleophile will be labeled. Mutagenic inactivation of a nucleophilic residue will result in loss of labeling if the residue in question is involved in forming a covalent phosphodiester bond with the DNA backbone. This type of experiment has been used to identify the nucleophilic, catalytic residue in the past (**Shuman et al., 1989**), and we sought to identify an alternative one using an analogous method. In the case of the wild-type heterodimer, both the GyrBA fusion and GyrA are labeled, consistent with both sides providing tyrosine as a nucleophile (**Figure 4**). Omitting $Mg^{2+}$ from the cleavage reaction drastically reduced the signal on both sides. With the heterodimer mutants, the signal is abolished on the side where the tyrosine is mutated, showing that tyrosine is absolutely required for cleavage and that no alternative nucleophile from the polypeptidic chain can substitute for it (**Figure 4**). The low amount of GyrA labeling observed in the $BA.A_F$ mutant is probably due to a very small amount of contaminating wild-type GyrA from the endogenous (chromosomal) *gyrA* gene in the *E. coli* expression strain. There is no background labeling at all for the GyrBA fusion since the coding sequence is absent in *E. coli*. This demonstrates that there is no nucleophile from the polypeptidic chain that can substitute for tyrosine.

However, there is a possibility that the nucleophile comes from the solvent in the form of a water molecule. If this was the case, the 5' end of the cleaved DNA would be exposed to exonuclease treatment and would not be protected by the normally covalently bonded polypeptide. We therefore

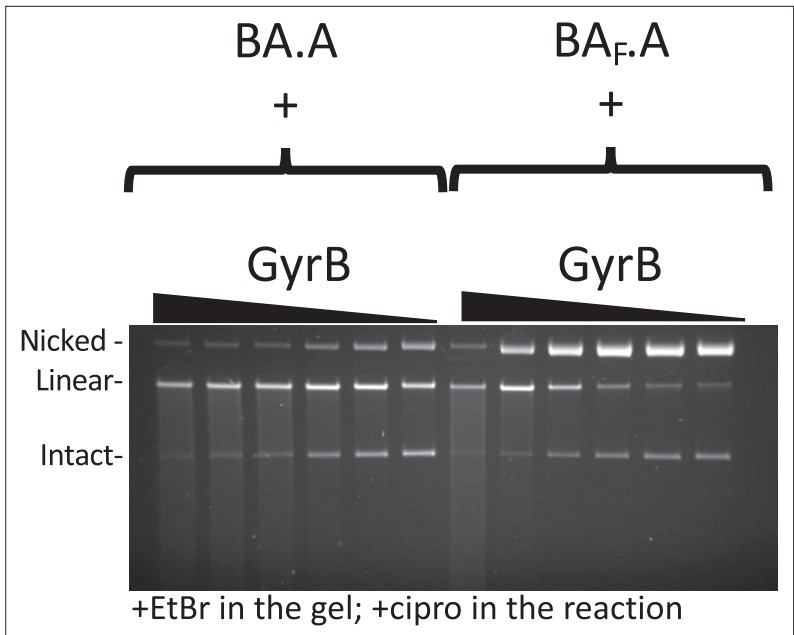

**Figure 5.** Effect of GyrB dose on subunit exchange. Five pmole of heterodimers (BA.A and BA$_F$.A as indicated) were incubated with increasing amounts of GyrB (triangle) in a cleavage assay. From the highest to the lowest dose: 16, 8, 4, 2, 1, and 0.5 pmole were used. At lower GyrB/heterodimer ratios, single-strand cleavage is predominant with BA$_F$.A, whereas BA.A still displays robust double-strand cleavage activity. Single-strand cleavage activity does go up marginally with the wild-type protein at the lower GyrB dose, suggesting missing GyrB on one side can lead to the formation of single-strand cleavage complexes. However, even at very low GyrB/heterodimer ratios, the majority of cleavage complexes are double-stranded.

performed a cleavage assay and compared the sensitivity of the cleaved products to T5 exonuclease, which requires a 5′ DNA end to degrade DNA, and Exonuclease III which requires a 3′ end (***Figure 4—figure supplement 1***). In both cases, we added a DNA PCR fragment to control for nuclease activity. Both nucleases were unable to digest intact plasmidic DNA as expected but were able to digest the control DNA fragment. Exo III was able to digest the gyrase cleavage product since the covalent link is established on the 5′ end, as expected. On the other hand, the gyrase cleavage products were resistant to T5 exo digestion, showing that the 5′ end of the cleaved product is protected by a covalently linked polypeptide, even when the tyrosine is mutated (***Figure 4—figure supplement 1***).

From all the experiments above, we suggest that the reconstitution of double-strand cleavage activity must come from interface swapping between two heterodimers.

## Free GyrB favors interface swapping

In all the experiments above, GyrB was added at approximately twofold excess compared to the heterodimers. At this concentration, we observe GyrB-dependent double-strand cleavage with BA.A$_F$. However, the only active product of whole subunit exchange (GyrBA fusion homodimer) for this construct should not require GyrB for its cleavage activity. We therefore interpret this result as GyrB promoting subunit exchange or at least DNA interface swapping, an additional activity to its classic role within a gyrase complex. We therefore tested the effect of varying GyrB concentration on the cleavage activity of the wild-type heterodimer (BA.A) compared to BA$_F$.A (***Figure 5***). In the case of the wild-type enzyme, robust double-strand cleavage activity at all GyrB concentrations was observed. Increasing the concentration of GyrB only marginally stimulated DSBs. Interestingly, robust double-strand cleavage activity was even observed at low GyrB/heterodimer ratio (less than one GyrB monomer for one heterodimer) suggesting that heterodimers alone can maintain a cleavage complex (although GyrB would still be needed for their formation, since they are not observed in the complete absence of GyrB). In the case of BA$_F$.A, adding GyrB at low concentration (GyrB/heterodimers <1) produced almost only single-strand broken cleavage complexes, whereas GyrB added at a high concentration (GyrB/heterodimers >1) produced the reconstitution of a robust double-strand

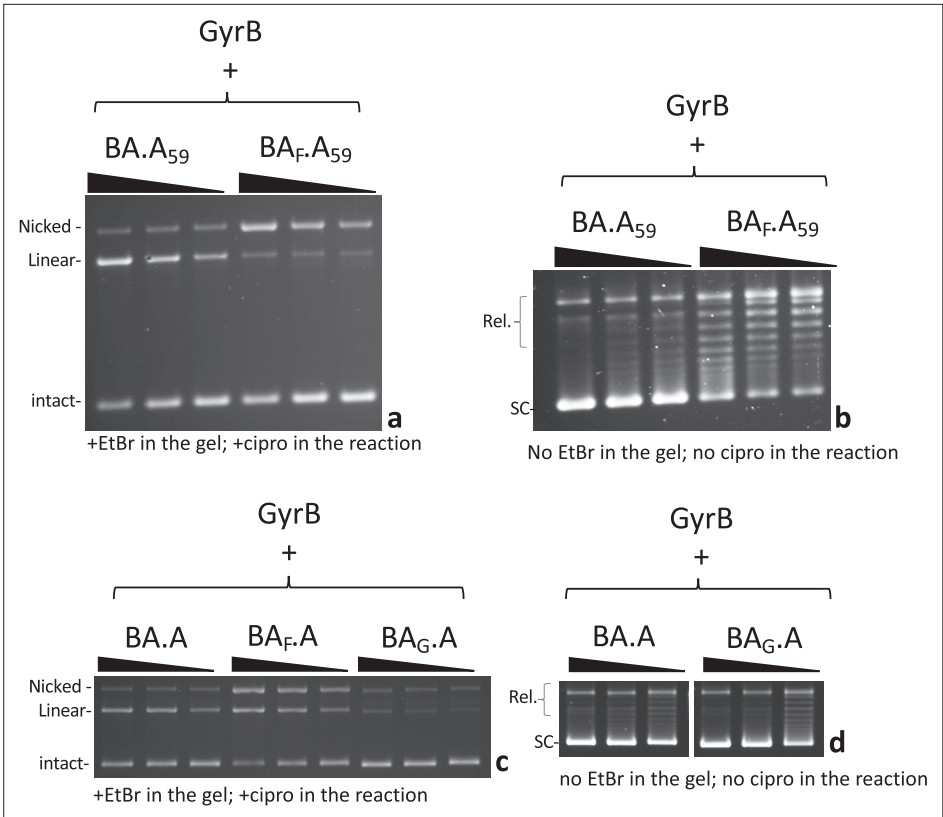

**Figure 6.** Effect of DNA wrapping and DNA bending defective mutants on subunit exchange. (**a**) 5, 2.5, and 1.25 pmole (triangle indicates increasing dose) of BA.A$_{59}$ and BA$_F$.A$_{59}$ (as indicated) were incubated in a cleavage assay with 8 pmole of GyrB. (**b**) 2.5, 1.25, and 0.625 pmole (triangle indicates increasing dose) of BA.A$_{59}$ and BA$_F$.A$_{59}$ (as indicated) were incubated in a supercoiling assay in the presence of 4 pmole of GyrB. (**c**) 5, 2.5, and 1.25 pmole (triangle indicates increasing dose) of BA.A, BA$_F$A and BA$_G$.A were incubated in a cleavage assay in the presence of 8 pmole of GyrB. (**d**) 2.5, 1.25, and 0.625 pmole (triangle indicates increasing dose) of BA.A$_{59}$ and BA$_F$.A$_{59}$ (as indicated) were incubated in a supercoiling assay in the presence of 4 pmole of GyrB. Omitting GyrB from all these assays abolished either the supercoiling or cleavage respectively. The exception being BA.A$_{59}$, which shows a small amount of double-strand cleavage in the absence of GyrB (*Figure 6—figure supplement 1*).

The online version of this article includes the following figure supplement(s) for figure 6:

**Figure supplement 1.** Cleavage and supercoiling activity of various heterodimer mutants.

**Figure supplement 2.** Analysis of cleavage and DNA binding properties of the GyrA-I174G mutant homodimer.

cleavage activity. This shows that our mutant heterodimer with a single catalytic tyrosine behaves as expected at low GyrB concentration and is indeed able to form single-strand cleavage complexes and further disproves the possibility of an initial GyrA dimer contamination of our preparations. In addition, this experiment suggests that the reconstitution of double-strand cleavage activity and therefore DNA gate interface swapping, is stimulated by a higher GyrB/heterodimer ratio.

## The CTD wrapping domain favor interface swapping

We next tested a construct where the CTD on the GyrA side of the heterodimer is deleted, abolishing the ability to wrap DNA on this side. This construct is dubbed A$_{59}$ (*Reece and Maxwell, 1991*), since the truncated GyrA is down to 59 kDa. We compared BA.A$_{59}$ to BA$_F$.A$_{59}$ (*Figure 6a and b*). The BA.A$_{59}$ enzyme displayed a robust double-strand cleavage activity and supercoiling activity, consistent with results that show only one wrapping domain is sufficient for supercoiling activity (*Stelljes et al., 2018*). The BA$_F$.A$_{59}$ mutant displayed mostly single-strand cleavage activity with minimal reconstitution of double-strand cleavage activity, in striking contrast with BA$_F$.A (at identical GyrB concentrations, and with the same GyrB preparation) (compare to *Figures 3 and 6a*). We therefore conclude that the wrapping domain favors interface swapping. The BA$_F$.A$_{59}$ mutant also showed reduced supercoiling activity

compared to BA.A$_{59}$ but interestingly it is not abolished. This observation is puzzling since interface swapping should not result in the reconstitution of an enzyme capable of supercoiling, since one side is missing a wrapping domain, while the other is missing a catalytic tyrosine. In our radiolabeling experiment, the BA.A$_F$ showed a very low labeling of the GyrA subunit (**Figure 4b**). This suggests that the BA.A$_F$ preparation is contaminated with a low amount of BA.A complexes where the GyrA subunit is encoded in the bacterial genome in the expression strain, and is therefore not mutated. The reverse complex BA$_F$.A does not show any contamination with an unmutated BA fusion subunit simply because the BA fusion is not encoded in the bacterial genome. We thus surmise that our BA$_F$.A$_{59}$ preparation is similarly contaminated with a very low amount of BA$_F$.A$_{endogenous}$ heterodimer, where the GyrA side is encoded by the endogenous *gyrA* gene. This GyrA would have the CTD and interface or subunit exchange would produce enough active GyrA dimer to detect supercoiling but insufficient amount to detect DNA cleavage (see Discussion and Supplementary discussion). We also compared BA$_F$.A$_{59}$ to BA$_{LLL}$.A$_{59}$ (**Figure 6—figure supplement 1a**); the LLL mutation favors interface swapping in the construct with both CTD domains present, and we found that it also favors reconstitution of double-strand cleavage activity when one CTD is absent (see Discussion for possible explanations, **Figure 6—figure supplement 1a**, compared with BA$_F$.A$_{59}$). Subunit exchange on the BA$_{LLL}$.A$_{59}$ should reconstitute an A$_{59}$ dimer, which is active for cleavage but cannot supercoil DNA. The double-strand cleavage observed with BA$_{LLL}$.A$_{59}$ did not correlate with an increase in supercoiling activity, compared with BA$_F$.A$_{59}$ (**Figure 6—figure supplement 1a**). This is consistent with the interpretation that the increased cleavage observed with BA$_{LLL}$.A$_{59}$ is due to the production of A$_{59}$ dimer by subunit exchange. In addition, this observation also further disproves the hypothesis that this double-strand cleavage is due to an alternative nucleophile in the BA$_{LLL}$.A$_{59}$ heterodimer since a heterodimer with double-strand cleavage activity and only one CTD (like BA.A$_{59}$) can supercoil DNA: if the BA$_{LLL}$.A$_{59}$ heterodimer had double-strand cleavage activity due to an alternative nucleophile, it is expected it could supercoil DNA, as it possess a CTD, unlike the A$_{59}$ dimers (see Supplementary discussion).

## DNA bending favors interface swapping

Next, we considered the effect of DNA binding itself on interface swapping. To do that we needed to affect DNA binding to the heterodimer, without affecting the GyrA side since DNA cleavage by GyrA dimers is our readout for DNA gate interface swapping. We therefore mutated Isoleucine 174, whose side chain 'pokes' between two bases of the bound G-segment DNA and introduce a 'kink' on each side. This severely bends the DNA that is being cleaved by the enzyme. It has been shown that mutating this residue abolished DNA bending, cleavage and strand passage activity in topoisomerase IV (**Lee et al., 2013**). When mutated to a glycine in *E. coli* gyrase, we found that the cleavage and supercoiling activity were abolished with dimeric GyrA$_{I174G}$ in the presence of GyrB (**Figure 6—figure supplement 2**). DNA binding was still present however (**Figure 6—figure supplement 2a**), consistent with earlier work (**Lee et al., 2013**). Moreover, we tested the ability of the mutant to wrap DNA by topological footprinting and found that the mutation did not abolish the wrapping, although it lowered its efficiency (**Figure 6—figure supplement 2c**). This suggests that DNA gyrase can wrap DNA prior to DNA bending. We introduced the I174G mutation on the GyrBA fusion side of the heterodimer (forming the BA$_G$.A construct) and tested the construct for cleavage. The BA$_G$.A dimer had much reduced single- and double-strand cleavage activity (**Figure 6c**). This result shows that abolishing DNA bending on one side is sufficient to impair cleavage on both sides. Moreover, it shows that only limited interface swapping is happening since it would result in the formation of wild-type GyrA dimers capable of double-strand cleavage. We do detect a small amount of double-strand cleavage though and the construct can introduce negative supercoils (**Figure 6d**). The BA$_G$.A heterodimer is not mutated for the catalytic tyrosine so it is possible that the bending induced by only one isoleucine is sufficient to produce the residual double-strand cleavage activity we observe; alternatively, interface swapping alone could explain this residual cleavage. These two explanations are not mutually exclusive.

## Kinetics of interface swapping

We have also examined the kinetics of double-strand cleavage with our various constructs. With the wild-type heterodimer (BA.A) cleavage complexes appear within 1–2 min and reach a maximum at 5–10 min. With BA$_F$.A and BA$_{LLL}$.A the appearance of cleavage complexes is delayed by a few minutes

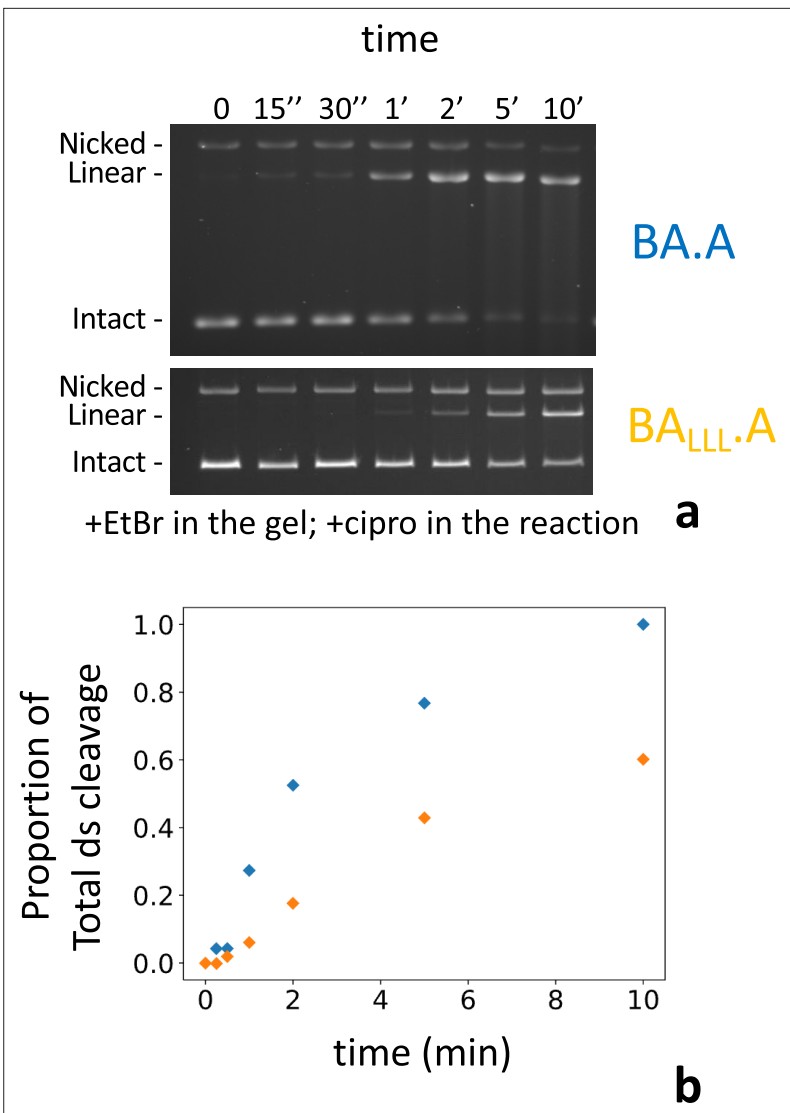

**Figure 7.** Kinetics of subunit exchange. (**a**) Five pmole of either BA.A (blue) or $BA_{LLL}$.A (orange) were incubated with 8 pmole of GyrB in a cleavage assay for the indicated times. $BA_{LLL}$.A +GyrB reaches a level of cleavage comparable to BA.A (albeit much later, around 60 min, not shown). (**b**) Quantification of the gels shown in (**a**). The proportion of linear of the total amount of material in each lane is plotted against time. The proportion of linear is normalized as the fraction of the total amount of linear reached at 60 min, minus background cleavage (low in that instance) observed at the 0 time point.

and starts appearing around 5 min and reach a maximum at 60 min (*Figure 7*). We show $BA_{LLL}$.A since the final amount of cleaved DNA is comparable to BA.A. This delay is therefore not due to a lower level of DNA cleavage (which could presumably take longer to rise above the detection level). We suggest that this delay corresponds to the time necessary for the exchange between interfaces to occur. Therefore, IS is rapid (around 3–5 min).

We have also looked at subunit exchange using BN-PAGE to try and directly observe the exchange product. We have performed BN-PAGE with various gyrase species (*Figure 2a* and *Figure 2—figure supplement 2a*), including GyrB, GyrA, $BA_F$.A, and A59. We have tried to mix GyrB with our heterodimer preparation to try and detect the product of exchange, without success. GyrB alone forms multimers (*Figure 2a* and *Figure 2a* and *Figure 2—figure supplement 2a*) and its addition to $BA_F$.A results in a complex that does not enter the gel.

We also looked at subunit exchange between a GyrA dimer and an $A_{59}$ dimer to try and observe the formation of an A.$A_{59}$ heterodimer. To identify the migration behavior of such a heterodimer,

we performed a denaturing/refolding procedure with a mix of GyrA and $A_{59}$ preparations. Denaturing disrupts the dimers and during renaturation dimerization occurs randomly with the available monomers in solution, thereby forming $A.A_{59}$ heterodimers (*Figure 2—figure supplement 2a and b*). The resulting pattern on BN-PAGE is complicated by the fact that both GyrA dimers and $A_{59}$ dimers can multimerize, which confers to the $A.A_{59}$ heterodimer the ability to multimerize as well. The bands obtained on BN-PAGE after denaturation and renaturation were identified by comparison with a native molecular weight marker and their reactivity to an antibody that recognizes GyrA, but not $A_{59}$ (*Figure 2—figure supplement 2c*). We assigned each band to their heterodimer/homodimer and multimerization status (*Figure 2—figure supplement 2b*). Mixing GyrA and $A_{59}$ does not immediately result in heterodimer formation without denaturation/renaturation, showing that subunit exchange usually does not happen rapidly with GyrA alone. This is consistent with the high stability of the interface. However, long incubation of the GyrA/$A_{59}$ mix did eventually result in the formation of heterodimers, but only after several days of incubation (*Figure 2—figure supplement 2d*). This happened without GyrB, ATP or ciprofloxacin. This slow exchange is distinct from our observed rapid reconstitution of double-strand activity which occurs in a few minutes and requires high concentration of GyrB. We conclude that the low concentration of GyrA monomer in solution (due to the stability of the interface) only allows for very slow subunit exchange between GyrA dimers and cannot account for our observation of a rapid reconstitution of double-strand cleavage activity from $BA_F.A$. In addition, this slow exchange was not affected by GyrB (*Figure 2—figure supplement 2*) and was partially inhibited by ATP and ciprofloxacin (*Figure 2—figure supplement 2d*). The rapid interface swapping described above depends on high concentration of GyrB and is unaffected by ciprofloxacin and ATP. It is also possible that only the DNA gate is exchanged, which would not result in the production of exchanged dimers (see Discussion).

## Discussion

In this work, we have purified stable heterodimers of GyrA and a GyrBA fusion construct (*Figure 2*). We have shown that mutating the catalytic tyrosine on one side of the heterodimer completely abolished cleavage on this side only (*Figure 4*). We observed the reconstitution of double-strand cleavage activity from a $BA_F.A$ and $BA.A_F$ mutated heterodimers with a mutated catalytic tyrosine on one side (*Figure 3*). This activity was dependent on the addition of a high concentration of GyrB (*Figure 5*). The appearance of this double-strand cleavage was delayed by a few minutes compared to the unmutated heterodimer (*Figure 7*). We conclude that gyrase can rapidly (within a few minutes) exchange its DNA gate sides in vitro, thereby allowing the mutated heterodimer to reconstitute double-strand cleavage activity and therefore supercoiling activity (*Figure 3*). We call this process 'interface swapping'. Interface swapping can result in whole subunit exchange depending on the number of interfaces involved (*Figures 1 and 8*, *Figure 1—figure supplement 2*).

We have carefully controlled our heterodimer preparation for contaminations that could explain these results. Our heterodimer preparations do not show any contamination by GyrA dimers either on BN-PAGE, by mass photometry or native mass spectrometry (*Figure 2*, *Figure 2—figure supplement 1*). The low contamination by either $(BA)_2$ or BA.A.B would produce double-strand cleavage independent of GyrB addition. This background cleavage is minimal (*Figure 3*). Therefore, we can confidently conclude that the cleavage reconstituted with the mutated $BA_F.A$ heterodimers does not arise from a contamination by GyrA dimers (see Supplementary discussion for the $BA.A_F$ complex).

The observation that heterodimeric single-sided catalytic mutant preparations (e.g. $BA.A_F$, $BA_F.A$, $BA_F.A_{59}$) still have some supercoiling activity, which we have confirmed, has led others (*Gubaev et al., 2016*) to propose a mechanism by which gyrase introduces negative supercoiling through nicking of the DNA (which requires only one catalytic tyrosine) and swivel relaxation of the DNA wrapped around the CTD domain of GyrA. We conclude that such a mechanism is most likely incorrect for several reasons. The first is a theoretical consideration. It is known that gyrase can introduce positive supercoiling by relaxing the constrained positive wrap around the CTD in the absence of ATP. Therefore, the default position of the CTD is the constraint of positive supercoils and their relaxation necessarily implies release from the CTD, which is favored by the ATPase activity of the enzyme (*Basu et al., 2018*; *Basu et al., 2012*). On average, +0.74–0.8 supercoils is constrained by wrapping around the CTD (*Basu et al., 2012*; *Kampranis et al., 1999*). The relaxation of a single positive loop (with a~+1 contribution to the ΔLk) should therefore introduce negative supercoiling by steps of 1. Only the

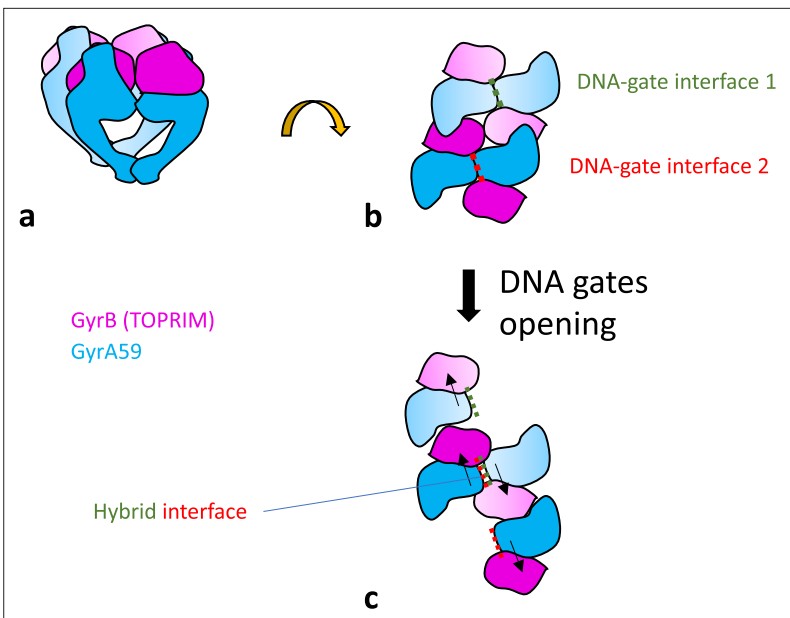

**Figure 8.** Proposed model for interface swapping. (**a**) Cartoon schematic of two gyrase complexes in close proximity, viewed from the side. Only the core complex is shown. (**b**) 90° rotation view. The gyrase 'super-complex' is viewed from the top. (**c**) The opening of each DNA-gate interface occurs by a sliding movement (black arrows), allowing reformation of a hybrid interface, in the center of the super-complex. The DNA is not represented.

The online version of this article includes the following figure supplement(s) for figure 8:

**Figure supplement 1.** Schematic of possible sequential interface swapping mechanism leading to complete subunit exchange for the $BA_F.A$ preparation.

**Figure supplement 2.** Schematic of possible sequential interface swapping mechanism leading to complete subunit exchange for the $BA.A_F$ preparation.

**Figure supplement 3.** Schematic of possible sequential interface swapping mechanism leading to complete subunit exchange for the $BA_F.A_{59}$ preparation.

**Figure supplement 4.** Schematic of possible sequential interface swapping mechanism leading to complete subunit exchange for the $BA_{59}.A_F$ preparation.

**Figure supplement 5.** Comparison of the activity of the GyrA dimer +free GyrB (wild type), BA.A heterodimer +free GyrB (heterodimer) and $(BA)_2$ (homodimer).

**Figure supplement 6.** Gyrase can stabilize cleavage complexes ~200 bp apart on DNA.

**Figure supplement 7.** Schematic of possible DNA gate-only interface swapping mechanism leading to a $BA_Q.A_F$ preparation having some supercoiling activity in the presence of GyrB.

conversion of a positive loop (+1) to a negative (–1) by strand passage can introduce supercoiling by steps of 2. Since a gyrase complex has two CTDs, it could be assumed that the simultaneous relaxation of the two loops could introduce supercoiling by step of two, however, Klostermeier and co-workers have proved that only one CTD is needed for introducing supercoils by steps of two (*Stelljes et al., 2018*). Therefore, for the model to stand, it must be assumed that, in addition to the (+1) supercoils stabilized by the writhe of the loop, another +1 must be stabilized in the twist of the double helix within this short loop. We find this to be unlikely considering recent cryo-EM structures of gyrase with DNA (*Vanden Broeck et al., 2019*). However, a CTD-wrapped state with +1.7 supercoils has been detected in the absence of ATP (*Basu et al., 2012*). It is therefore conceivable that the relaxation of DNA wrapped in this state could lead to the introduction of supercoiling by step of 2 via swiveling. This state is much less abundant than the +1 state however and the efficiency of the swiveling mechanism would be very low. For an efficient mechanism involving relaxation by swiveling of this +1.7 state one would have to assume that ATP favors its formation, in addition to promoting the release of the DNA loop. These two activities contradict each other. In addition, the addition of nucleotides either did not favor the +1.7 state or did not change its abundance (*Basu et al., 2018*). Finally, this state

could represent twin-wrapping around the 2 CTDs, and only one CTD is sufficient for supercoiling, as stated above. In conclusion, introduction of negative supercoiling by swiveling-mediated relaxation of the wrapped DNA loop would have to most likely occur by steps of 1, or be extremely inefficient when they occur by steps of two.

On top of the above consideration ours and others' (*Gubaev et al., 2016*) results can be explained by DNA gate interface swapping that reconstitute the double-strand cleavage activity that is necessary for negative supercoiling activity. *Figure 8—figure supplements 1–4*, *Figure 8—figure supplement 7* show schematics representing how the process of interface swapping and subunit exchange could happen for our various heterodimer constructions and some constructions reported elsewhere (*Gubaev et al., 2016*). This model allows some predictions about the abundance and activity of each species present in the prep, either initially or after IS and SE. These predictions are consistent with our experimental data and, we suggest, data reported by *Gubaev et al., 2016*.

In almost all heterodimer configurations, mutating one catalytic tyrosine results in an important diminution of supercoiling activity, accompanied by a drastic diminution of double-strand cleavage activity, suggesting that most of the activity is based on strand passage through a double-strand break. In a configuration where the reconstitution of double-strand cleavage activity remained important (such as BA$_F$.A, *Figure 8—figure supplement 1*), it correlates with a robust supercoiling activity, further supporting that supercoiling activity is based on the ability to cleave DNA in both strands. These results argue strongly against swiveling since the later only requires a single-strand break, and therefore should be unaffected by mutating one catalytic tyrosine.

A series of experiments (*Gubaev et al., 2016*) were interpreted as excluding subunit exchange and interface swapping (the authors do not distinguish between the two). Namely the absence of double-strand cleavage observed with heterodimers and FRET experiments interpreting the absence of FRET as showing that the DNA gates of different heterodimer never come into close proximity. We argue that these experiments do not fully exclude SE or IS. Negative supercoiling by DNA gyrase is an extremely efficient reaction and this activity can be detected with a very low concentration of the active enzyme. Consequently, it is possible to detect efficient negative supercoiling and not be able to detect the double-strand cleavage from which the activity originates because of the low concentration of active enzyme. We argue that DNA gate IS that reconstitute double-strand cleavage activity occurs but is not detected by cleavage assays or FRET (*Gubaev et al., 2016*) because of the low proportion of complexes where IS has occurred. Despite the low number of exchanged complexes able to double-strand cleave DNA, negative supercoiling can be observed because of the high efficiency of the reaction.

We have shown that efficient double-strand cleavage reconstitution by DNA gate IS only occurs in specific conditions, namely with the BA$_F$.A complex in the presence of an excess of GyrB. In *Gubaev et al., 2016*, cleavage experiments are only shown for BA.A$_F$, and display low double-strand cleavage. This agrees with our experiments that show barely detectable double-strand cleavage with the BA.A$_F$ complex (*Figure 8—figure supplement 8–supp. and 2*). Considering that a low amount of DNA gate IS can explain results shown in Gubaev et al., more experiments are needed to exclude it. For instance, cleavage assays with the heterodimer are shown (*Gubaev et al., 2016*), and we argue that there is a very slight increase in double-strand breaks observed with BA.A$_F$ +GyrB and not observed in the absence of GyrB. More experiments could have been done to investigate this, namely increase the GyrB concentration (it is known that GyrB favors cleavage) and purify the cleavage complexes. We have done this with the BA.A$_F$ and shown unambiguously that a low amount of DNA gate IS occurs (see below for explanation of the difference between the two types of heterodimers).

In addition, the supercoiling activity seen with the heterodimer is delayed by several minutes compared to the wild-type homodimer (*Gubaev et al., 2016*). This is completely consistent with our experiment showing a delay of several minutes in the appearance of double-strand cleavage with the heterodimer (*Figure 7*). If the wild-type enzyme and the heterodimer use the same swiveling mechanism, why is this delay observed? Another way to ask the same question would be: if gyrase uses a swiveling mechanism that only uses one catalytic tyrosine, why does the efficiency of supercoiling go down when a single tyrosine is mutated on one side? This is not addressed (*Gubaev et al., 2016*); we suggest that the efficiency of supercoiling by heterodimers will be correlated to the efficiency of double-strand cleavage reconstitution. Consistently, complexes with a higher level of reconstituted double-strand cleavage (BA$_F$.A) have a higher supercoiling efficiency.

We have observed that heterodimers that should be incapable of reconstituting an active gyrase when the whole subunits are exchanged, namely $BA_F.A_{59}$, is still able to reconstitute a low supercoiling activity (*Figure 6*). We have interpreted this result by the observed low contamination by $BA_F$. $A_{endogenous}$, which undergoes subunit exchange (*Figure 8—figure supplement 8–supp. and 3*). Elsewhere it is reported that the reverse complex, $BA_{59}.A_F$ (called $BA_{\Delta CTD}.A_F$) can reconstitute supercoiling activity (*Gubaev et al., 2016*). However, the observed supercoiling is very low, much lower than the other heterodimers and the reason for this is not discussed. Such an amount of supercoiling could be explained by even the most minimal contamination by an active double-strand cleaving gyrase and therefore extreme caution should be used to interpret these data. Deleting the CTD on one side when two tyrosines are present does not reduces supercoiling activity as much and therefore, at the very least, one should conclude that the swiveling mechanism is much less efficient than the strand-passage mechanism which requires requires two tyrosines. We argue that $BA_{59}.A_F$ (*Gubaev et al., 2016*) could also be contaminated with $BA_{59}.A_{endogenous}$, which would have a lower efficiency of subunit exchange than $BA_F.A_{endogenous}$ (this study) since they lack a CTD on the BA fusion side (see Supplementary discussion and *Figure 8—figure supplement 4*). This would explain why the supercoiling activity of $BA_{59}.A_F$ is so low. The final possibility would be that the exchanged DNA gate could use the CTD of the neighboring complex for wrapping. However, our $BA_{LLL}.A_{59}$ shows that it is not the case. $BA_{LLL}$. $A_{59}$ shows higher reconstitution of double-strand cleavage through interface swapping. The reconstituted active DNA gate is therefore between the two $A_{59}$ subunits and if this DNA gate could use the CTD from the neighboring BA fusion, we should expect this extra double-strand cleavage to produce negative supercoiling activity. The extra IS observed with $BA_{LLL}.A_{59}$ is not accompanied by supercoiling activity (*Figure 6—figure supplement 1a*) and therefore the data shows that an active DNA gate can only use a CTD from the same GyrA subunit for strand passage and negative supercoiling. We conclude that the low supercoiling reconstituted by $BA_F.A_{59}$ (this study) and $BA_{59}.A_F$ (*Gubaev et al., 2016*) are likely to originate from low amount of contamination (see Supplementary discussion).

The reasons for the asymmetry between $BA_F.A$ and $BA.A_F$ are unclear so far. With the $BA_F.A$ heterodimer subunit exchange produces an active GyrA dimer and an inactive $(BA_F)_2$ dimer. In the case of $BA.A_F$, subunit exchange reconstitutes an active $(BA)_2$ dimer and an inactive $GyrA_F$ dimer. Therefore, the difference in the level of cleavage and supercoiling activity reconstituted could stem from a difference in activity between the GyrA dimer (with free GyrB) and the $(BA)_2$ dimer. We have directly tested this and compared both cleavage and supercoiling activity between GyrA dimer ('wild type'), $(BA)_2$ dimer and BA.A heterodimer (*Figure 8—figure supplement 5*). We found that the activity of $(BA)_2$ is reduced by a factor 100. Interestingly, the heterodimer has a level of activity intermediate between the GyrA dimer and the $(BA)_2$ dimer. This is true for both cleavage and supercoiling. These results can explain the asymmetry between $BA_F.A$ and $BA.A_F$ since the subunit exchange with the former produces a more active complex by subunit exchange compared to the later. These results also show that fusing GyrB to GyrA limits both cleavage and supercoiling activity. We did not add a linker to our fusion and therefore the position of GyrB with respect to GyrA and DNA is expected to be more constrained than with free GyrB. It has been reported that the GyrB dimer is leaning to one side with respect to the dyad axis in the gyrase-DNA complex (*Vanden Broeck et al., 2019*). Therefore, in this complex, the position of each GyrB subunit with regard to GyrA is different and this flexibility of GyrB in the gyrase complex might be important for its activity. We speculate that with a GyrB fused to GyrA limits the leaning ability of the GyrB dimer and therefore reduces both the supercoiling activity and the ability to form cleavage complexes.

This leads us to discuss the exact mechanism of interface swapping and subunit exchange. It is a priori a surprising result considering the stability of the gyrase dyad interface and its importance in ensuring genome stability. However, during strand passage all three interfaces must be broken and the opening of the interfaces necessarily implies the possibility of exchange between two complexes prior to their reformation. The integrity of the complex is ensured through sequential opening and closing of the three interfaces. This sequential opening and closing allows the transported DNA to go through the whole complex without it coming apart. Similarly, subunit exchange could also occur sequentially, one interface after the other, allowing the partial or whole exchange to occur without the complex coming apart (*Figure 1b*, *Figure 1—figure supplement 2*). This model would only require gyrase complexes being able to come together and interact in an appropriate configuration and we indeed observe that both GyrA dimers and GyrB monomers can multimerize, each on their own in

solution, supporting this hypothesis. In fact, our results only show that the DNA-cleaving interface is exchanged (DNA gate IS); it is possible that this is not followed by the exchange of the C-gate interface, producing a sort of helical super-complex where one interface interacts with a different subunit compared to the other interface (*Figure 1—figure supplement 2*, *Figure 8 – Figure 7*). Structural evidence exists for such a complex. In earlier work (*Rudolph and Klostermeier, 2013*) a GyrA structure with the DNA gate opened was obtained. The DNA gate interacts with the next GyrA subunit forming a helical oligomer of GyrA within the crystal. We suggest that this structure could underly IS in our system, with the C-gate interface remaining closed, while the DNA gate interface opens and reforms with the neighboring complex. It is even possible that such a super-complex would retain negative supercoiling activity. The opening of the DNA gate does not occur by the pulling apart of the GyrA WHD domains but rather by a sliding of the two GyrA domains against one another along an axis close to being perpendicular to a plane formed by the dyad axis and the DNA axis (*Chen et al., 2018*; *Germe et al., 2018*; *Wendorff et al., 2012*). Therefore, the freed DNA-gate interface could be available to dimerize with a neighboring interface similarly freed by the opening of the DNA gate. The only requirement being that the neighboring complex is positioned parallel to the other one, in such a way that the simultaneous opening of the DNA gates results in the meeting of the two DNA gate interfaces between the complexes, thereby achieving interface swapping (*Figure 8*, *Figure 1—figure supplement 2*). If within this complex only the DNA-gate is swapped with a neighboring heterodimer, one would expect the reconstitution of an active super-complex where the swapped DNA-gate reconstitutes double-strand DNA cleavage while the unswapped ATPase domains retain strand capture activity (see Supplementary discussion).

We have tried and failed to directly observe subunit exchange by mass photometry. We tried to observe the reconstitution of a GyrA dimer from a heterodimer. However, this experiment is complicated because of several factors. The first is the necessity of a high concentration of GyrB. This will mask any GyrA monomer that might be produced and will bind any GyrA dimer that is produced reconstituting a trimer (with a MW equivalent to the heterodimer) or a tetramer (with a MW equivalent to GyrB bound to a heterodimer) that are indistinguishable from the original complex. We have not observed a significant change of the mass photometry profile of a mixture of GyrB +BA .A heterodimer over time. In addition, our data suggest that proper DNA binding and bending is necessary for interface swapping. Adding a long piece of DNA only scrambles the mass photometry profile as the complexes binds to it to form high MW particles. We have tried to add a short, purified DNA fragment (170 bp) which gives a nice peak on its own in mass photometry but again failed to see a difference. It could be that longer DNA is needed to allow the wrapping of several CTDs along the DNA and position the complexes for subunit exchange (see below). It is also possible that only the DNA-gate interfaces are exchanged and therefore no GyrA dimers are ever produced, only higher MW oligomers that are difficult to detect. Indeed, interface swapping does not necessarily result in the coming apart and complete separation of the whole complex (see above). Therefore, due to its partial nature, complexes with swapped interfaces are likely difficult to observe. This could further account for the lack of subunit exchange observed by FRET. FRET can only be observed on a stable complex, and we note that it was tested either before or after the supercoiling reaction not during the reaction when they would occur transiently. The only condition that showed stable, exchanged interfaces is with the $BA_F.A$ +GyrB reaction in the presence of ciprofloxacin which produces stable, reversible double-strand cleavage. We surmise that cross-linking the DNA interface in these could provide direct evidence of interface swapping. *Gubaev et al., 2016* have done no cross-linking experiments; these experiments are not straightforward since they require specific cross-linking at a functional DNA gate and a general cross-linker cannot be used.

We have shown that efficient IS requires the CTD or wrapping domain of GyrA. IS is quick in our system and removing the CTD produced much less double-strand cleavage within the 30 min timeframe of our experiment. We suggest that the wrapping of the DNA is necessary for the positioning of two or several gyrase complexes in such a way that interface exchange by the mechanism proposed above is favored. It is also possible that the CTD favors the opening of the DNA-gate interface, an idea supported by the observation that full-length gyrase has ATP-independent strand passage activity but not the $A_{59}$ dimer +GyrB.

We also showed that mutating isoleucine 174 drastically diminished the amount of IS. This could be interpreted in various ways. First, proper DNA bending is necessary for DNA cleavage (*Dong and*

*Berger, 2007*; *Lee et al., 2013*), and X-ray structures have shown that DNA binding and bending correlate with a slight opening of the DNA gate (*Germe et al., 2018*; *Wendorff et al., 2012*). Therefore, it could be that the binding of the DNA loosens the DNA gate interface and favors its opening. The second interpretation is that, although the isoleucine mutant can still bind and wrap DNA, the wrapping efficiency is reduced and therefore indirectly affects interface swapping since the CTD is important for IS.

We also observed that mutating the two residues neighboring the catalytic tyrosine favored IS. The mutation converts polar side chains to aliphatic ones and likely affects interface interaction. Indeed, the catalytic tyrosine and its neighboring residues are in proximity to the TOPRIM domain in the gyrase apo structure (*Bax et al., 2019*). Upon binding DNA, the distance between them increases and DNA cleavage is accompanied by the full disengagement of the catalytic tyrosine from the TOPRIM domain. The opening of the DNA gate fully separates the catalytic loop, which contains the catalytic tyrosine, from the TOPRIM domain by a sliding motion. Therefore, mutations disfavoring interaction between the catalytic loop and the TOPRIM domain are likely to favor DNA gate opening and therefore subunit exchange.

Subunit exchange, which necessitates IS, has been proposed as the mechanism behind DNA gyrase-mediated illegitimate recombination (*Ikeda et al., 2004*). Gyrase was shown to be involved in the transfer of an antibiotic resistance cassette from a plasmid to lambda DNA in a packaging reaction, in the presence of a quinolone (*Ikeda et al., 1982*; *Ikeda et al., 1981*). This result was independent of any homologous recombination pathway in either the phage or bacterium (*Ikeda et al., 1982*; *Ikeda et al., 1981*). The authors suggested that gyrase-mediated recombination in one of two ways: either (1) by dissolution of the gyrase complex into heterodimers (GyrA:GyrB) that still had the DNA bound in the phosphotyrosyl bond, which could reassociate with another GyrA:GyrB heterodimer in a similar situation, allowing for recombination, or (2) by the formation of higher order oligomers (GyrA$_4$GyrB$_4$) and interface swapping, which upon dissolution of the higher-order complex could result in subunit exchange and recombination (*Ikeda, 1994*; *Ikeda et al., 2004*). We suggest the former may be the cause of slower subunit exchange observed in the formation of GyrA and GyrA59 heterodimers over several days (*Figure 2—figure supplement 2*). This slow subunit exchange is inhibited by DNA and ciprofloxacin in contrast with the rapid subunit exchange observed with heterodimers. Therefore, the two are mechanistically distinct. Indeed, solution-based slow subunit exchange requires the complete separation of the complex for the formation of free monomers but does not require the formation of higher-order oligomers. In contrast, rapid subunit exchange by interface swapping does not require the complete separation of the complex but does involve the formation of higher-order oligomers. It therefore makes sense that one, and not the other would be inhibited by DNA and ciprofloxacin. Ciprofloxacin-induced cleavage complexes do not come apart in solution, which is consistent with ciprofloxacin inhibiting the formation of free monomers. Ciprofloxacin does not prevent rapid interface swapping and it has been shown that it does not prevent DNA-gate opening (*Chan et al., 2017*). Finally, while deleting the CTD affects rapid subunit exchange, it does not prevent, slow, solution-based exchange (*Figure 2—figure supplement 2*). We surmise that illegitimate recombination may occur within higher-order oligomer formation followed by interface swapping (the DNA gate specifically). This is consistent with the timescale of illegitimate recombination and explains how two DNA duplexes are brought together for recombination by two gyrase complexes.

Considering the ubiquity and abundance of gyrase in bacterial cells, the rapid interface swapping mechanism is expected to occur in vivo where DNA is present and gyrase is highly concentrated. If interfaces are swapped when gyrase is covalently bound to cleaved DNA on both sides, followed by re-ligation with the swapped DNA-gate, this would result in illegitimate recombination. The way the two gyrase complexes interact would bring together two DNA duplexes for recombination. The presence of fluoroquinolones, by extending the lifespan of the cleaved state, would make this process much more likely. We have observed that GyrB on its own can oligomerize in a regular fashion where the intensity of the dimer, trimer, tetramer and so forth diminishes following roughly an exponential until the oligomers can no longer be resolved (carefully consider *Figure 2—figure supplement 2a*, lane B). This observation strongly suggests a regular oligomerization and tentatively a linear oligomer of a GyrB monomer repetition. We have observed that a high gyrase/DNA ratio in the presence of ciprofloxacin produces cleavage complexes ~200 bp apart (*Figure 8—figure supplement 6*). Considering that gyrase can interact with ~150 bp of DNA through DNA wrapping, it suggests that the DNA

exiting a gyrase complex is immediately engaged in another neighboring gyrase-DNA complex. This linear repetition of cleavage complexes on DNA (spanning the whole plasmid in this instance) mirror the linear repetition of GyrB monomers within GyrB oligomers. We suggest that GyrB can drive the close repetition of many cleavage complexes on linear DNA. Again, the high concentration of gyrase in bacterial cells is consistent with the possibility of a long-range linear gyrase oligomer forming on the genome of bacteria. The tight packing of gyrase, observed on plasmids, and potentially driven by GyrB oligomerization, therefore opens the possibility of micro-deletions or -inversions by inter-face swapping of the cleavage complexes on the bacterial genome, induced by fluoroquinolones. The necessary re-ligation can happen if the compound is washed-out, and we speculate that tempo-rary exposure to high dose of fluoroquinolones could induce these rearrangements. Longer-range genomic rearrangements are also conceivable depending on the organization of gyrase clusters in the bacterial cell. This interface swapping mechanism could therefore have profound consequences for antibiotics treatment of infections and more generally the evolution of the bacterial lineage, within which DNA gyrase is ubiquitous.

## Materials and methods

### Expression and purification of GyrA-GyrBA heterodimer, GyrB and GyrA

The full *E. coli* GyrA coding sequence was cloned in pACYC-Duet (from Millipore/Novagen) under the T7 promoter. A fusion of *E. coli* gyrB followed by *E. coli* gyrA, without linker was cloned into pTTQ18, under the tac promoter. A 6xHis tag is added at the N-terminus of GyrB. Both plasmids were co-transformed into *E. coli* BLR competent cells (Millipore/Novagen). Expression was initiated by the addition of 1 mM IPTG to 1 L of log-phase culture in LB and allowed to proceed for 3 hr at 37°C. Cells were then pelleted and kept at –80 ° C until the purification procedure was performed. Expression of GyrBA is much lower than GyrA and ensures that only a low amount of GyrBA fusion homodimers is formed (*Figure 1—figure supplement 1*). For purification the cells were resuspended in 10 mL of 50 mM Tris·HCl (pH 7.5), 10% sucrose, 10 mM KCl, 1 mM EDTA, 2 mM DTT buffer to which was added 1 tablet of complete EDTA-free protease inhibitors cocktail (Roche). The cells were lysed by passage through an Avestin high pressure system at 45 psi. The lysate was cleared by centrifugation and loaded on to a 5 ml Ni²⁺ column at 0.5 mL/min in 50 mM Tris·HCl (pH 7.5), 1 mM EDTA, 2 mM DTT, 20 mM imidazole. Elution proceeded in a 30 mL gradient to 500 mM imidazole. The heterodimer fractions were collected, concentrated and loaded onto a Superose 6 column and eluted in 50 mM Tris·HCl (pH 7.5), 1 mM EDTA, 2 mM DTT. The heterodimers fractions were pooled and concentrated to around 2 mg/mL. Glycerol (to 10%) and KCl (to 100 mM) were added, and the preparation frozen in small aliquots in liquid $N_2$ and kept at –80 °C. A monoQ step can be added before the Ni²⁺ column and this increases the purity of the preparation and lowers background cleavage. However, results are similar. The individual gyrase subunits GyrA and GyrB were prepared as described previously (*Maxwell and Howells, 1999*).

### Cleavage assays

Cleavage reactions were performed as described previously (*Reece and Maxwell, 1989*) with some modifications. 500 ng of relaxed pBR322 plasmid is used in each 30 µL reaction containing 35 mM Tris·HCl (pH 7.5), 24 mM KCl, 4 mM $MgCl_2$, 2 mM DTT, 6.5% (w/v) glycerol, and 0.1 mg/mL albumin. The reactions were carried out for 30 min (unless indicated otherwise) at 37 °C. The amount of gyrase preparation added to each reaction is indicated for each experiment. Cleavage complexes were trapped by the addition of 7.5 µL of 1% (w/v) SDS. The cleavage products were then digested with 20 µg of proteinase K for 1 hr at 37 °C. The cleaved DNAs were analyzed using 1% agarose elec-trophoresis in the presence 0.5 µg/ml ethidium bromide. In these conditions, intact DNA plasmids migrate as a single band (topoisomers are not resolved). Gels were quantified with imageJ and the data treated and plotted with Scipy (*Jones et al., 2001*).

### Supercoiling assays

Supercoiling reactions were performed as described previously (*Reece and Maxwell, 1989*). A total of 500 ng of relaxed pBR322 plasmid was used as a substrate in a 30 µL reaction containing 35 mM

Tris·HCl (pH 7.5), 24 mM KCl, 4 mM MgCl$_2$, 2 mM DTT, 6.5% (w/v) glycerol, 1.8 mM spermidine, 1 mM ATP and 0.1 mg/mL albumin. The reactions were carried out for 30 min at 37 °C. Topoisomers were resolved by 1% agarose gel electrophoresis in the absence of ethidium bromide. The amount of enzyme added varies and is indicated for each experiment. It is usually kept limiting to compare different preparations. The efficiency of supercoiling can vary depending on the assay buffer and GyrB preparation used. In this manuscript, assays presented in the same figure (supplementary and main) are all done with the same buffer and GyrB preparation.

## Blue-native PAGE

Proteins (normally between 0.1 µg and 2 µg) were added, along with other relevant factors, such as ATP, DNA and or antibiotics, native-PAGE sample buffer (2.5 µL of a 4×solution - 50 mM Bis-Tris (pH 7.2), 6 M HCl, 50 mM NaCl, 10% (w/v) glycerol, and 0.001% Ponceau S in Milli-Q Ultrapure H$_2$O) and ultrapure water (to a final volume of 10 µL) to a 10 or 15-well NativePAGE Novex 4–16% gradient Bis-Tris Gel (Life Technologies). Gels were placed in a XCell SureLock Mini-Cell Electrophoresis System in the cold room (~7 °C). The upper buffer chamber was filled with cold 1×Cathode buffer (50 mM Tricine, 15 mM Bis-Tris and 0.002% Coomassie G250 in Milli-Q Ultrapure H$_2$O, pH 7.0) and the lower chamber filled with cold 1×Anode buffer (50 mM Bis-Tris (pH 7.0) in Milli-Q Ultrapure H$_2$O). The gels were run at 150 V, restricted to 8 mA (max 10 mA for 2 gels) for 1 hr, after which the voltage was increased to 250 V, restricted to 200 mA per gel for a further 2 hr. The gels were either stained with InstantBlue Coomassie (Expedeon) stain or transferred for Western Blotting. Gel pictures were taken using a Syngene G:BOX Gel Doc system.

## Western blot for BN-PAGE

Polyacrylamide gels were transferred to PVDF membranes using the BioRad Trans-Blot Turbo system at 20 V, 2.5 mA, for 15 min. After transfer, membranes were briefly washed with Ponceau S then rinsed with ultrapure MilliQ H$_2$O. The membrane was blocked in TBS-T (50 mM Tris.HCl pH 7.6, 150 mM NaCl, 0.1% Tween-20) with 5% milk solids (Marvel Dry Skimmed Milk powder) for 10 min before incubating at 4 °C overnight with monoclonal antibody (either anti-GyrA-CTD – 4D3 or anti-GyrB-CTD – 9G8; a gift from Alison Howells, Inspiralis) diluted 1/1000 in TBS-T 5% milk. The membrane was then rinsed briefly with TBS-T before washing three times for 10 min each at room temperature. The membrane was then incubated at room temperature for 1 hr with secondary antibody (1/5000); rabbit polyclonal antimouse-HRP conjugate (Dako). This was then washed as described above. The membrane was flooded with Pierce ECL Western Blotting Substrate and left for 1 min at room temperature before covering with Clingfilm and exposed for 5–10 min onto Amersham Hyperfilm ECL Auto Radiography film before developing in a Konica Minolta SRX101A developer.

## Mass photometry

We performed measurements using the first generation Refeyn instrument. The protein preparations were diluted to the optimal concentration (as determined through trial-and-error) in the cleavage reaction buffer used for cleavage assays. Ten µL of buffer was deposited on a glass slide cleaned by sonication in isopropanol. Focus was established automatically by the instrument and 1–2 µl of the diluted preparation was added and mixed rapidly by pipetting. A 6000-frame movie of the collisions was recorded and analyzed with the Refeyn software (AcquireMP and DiscoverMP, respectively). The contrast of the landing events on the surface (point spread functions) are analyzed and converted to a molecular mass by using a calibrant (Urease) that has well-defined molecular mass peaks (91, 272, and 545 kDa). The number of collisions is plotted against the measured molecular weight and the peaks were fitted to a gaussian function using DiscoverMP.

## Cleavage-induced radiolabeling of heterodimeric gyrase

Relaxed pBR322 was radiolabeled using a commercial nick-translation kit and α$^{32}$P-dCTP. Cleavage assays were then performed as above replacing the DNA substrate with 250 ng of radiolabeled DNA. Around 10 pmole of heterodimers were used for each reaction and following SDS trapping, no proteinase K was added. 150 mM of NaCl was added along with 10 µg of albumin and the cleavage complexes were precipitated by addition of 2 volumes of absolute ethanol. The pellets

were resuspended in 20 µl water and digested with micrococcal nuclease. The labelled proteins were analysed by SDS-PAGE. Following migration, the gel was dried and exposed to a phosphor screen.

## Native mass spectrometry

The protein preparation was stored at −20 °C before being buffer exchanged into 200 mM ammonium acetate (pH 6.8) by multiple rounds of concentration and dilution using the Pierce protein concentrators (Thermo Fisher). The sample was then diluted to 2 µM monomer concentration immediately before the measurements. The data was collected using in-house gold-plated capillaries on a Q Exactive mass spectrometer in positive ion mode with a source temperature of 50 °C and a capillary voltage of 1.2 kV. For GyrA, in-source trapping was set to −200 V to help with the dissociation of small ion adducts and HCD voltage was set to 200 V. For $BA_F.A$, the voltages were set to −100 V and −100 V for in-source trapping and HCD, respectively. Ion transfer optics and voltage gradient throughout the instruments were optimized for ideal transmission. Spectra were acquired with 10 microscans to increase the signal-to-noise ratio with transient times of 64ms, corresponding to the resolution of 17,500 at $m/z=200$, and AGC target of $1.0\times10^6$. The noise threshold parameter was set to 3 and variable m/z scan ranges were used.

## Acknowledgements

We thank James Berger for critical reading of the manuscript. This work was supported by a Wellcome Trust Investigator Award (110072/Z/15/Z) and a Biotechnology and Biosciences Research Council (UK) Institute Strategic Programme Grant (BB/P012523/1),

## Additional information

### Funding

| Funder | Grant reference number | Author |
|---|---|---|
| Wellcome Trust | 10.35802/110072 | Thomas RM Germe<br>Natassja G Bush<br>Victoria M Baskerville<br>Dominik Saman<br>Justin LP Benesch<br>Anthony Maxwell |
| Biotechnology and Biological Sciences Research Council | BB/P012523/1 | Thomas RM Germe<br>Natassja G Bush<br>Victoria M Baskerville<br>Dominik Saman<br>Justin LP Benesch<br>Anthony Maxwell |

The funders had no role in study design, data collection and interpretation, or the decision to submit the work for publication. For the purpose of Open Access, the authors have applied a CC BY public copyright license to any Author Accepted Manuscript version arising from this submission.

### Author contributions

Thomas RM Germe, Conceptualization, Formal analysis, Supervision, Validation, Investigation, Visualization, Methodology, Writing – original draft, Project administration, Writing – review and editing; Natassja G Bush, Investigation, Methodology, Writing – review and editing; Victoria M Baskerville, Visualization, Methodology; Dominik Saman, Investigation, Visualization, Methodology; Justin LP Benesch, Funding acquisition, Methodology; Anthony Maxwell, Conceptualization, Supervision, Funding acquisition, Investigation, Methodology, Writing – review and editing

### Author ORCIDs

Thomas RM Germe https://orcid.org/0000-0002-6885-0971
Natassja G Bush https://orcid.org/0000-0003-3479-1261
Justin LP Benesch https://orcid.org/0000-0002-1507-3742
Anthony Maxwell http://orcid.org/0000-0002-5756-6430

Reviewer #1 (Public Review): https://doi.org/10.7554/eLife.86722.3.sa1
Reviewer #2 (Public Review): https://doi.org/10.7554/eLife.86722.3.sa2
Author response https://doi.org/10.7554/eLife.86722.3.sa3

## Additional files

### Supplementary files
• MDAR checklist

### Data availability

All the raw data are freely available at https://doi.org/10.5061/dryad.w6m905qwn. Reagents are freely available by writing to thomas.germe@cuanschutz.edu.

The following dataset was generated:

| Author(s) | Year | Dataset title | Dataset URL | Database and Identifier |
|---|---|---|---|---|
| Germe T | 2024 | Rapid, DNA-induced interface swapping by DNA gyrase | https://doi.org/10.5061/dryad.w6m905qwn | Dryad Digital Repository, 10.5061/dryad.w6m905qwn |

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

## Appendix 1

### Supplementary discussion

#### Low contamination of BA.A$_F$ and BA$_F$.A$_{59}$ with endogenous GyrA

The reconstitution of double-strand cleavage correlates with the reconstitution of supercoiling activity. However, since the supercoiling reaction is extremely efficient, as opposed to cleavage, a very low amount of contaminating GyrA dimers, below detection levels, could explain the detection of low level of negative supercoiling (see Discussion). In the case of BA$_F$.A the level of supercoiling is very high, comparable to the wild-type, and cannot be explained by a sub-detection level of GyrA. However, In the case of BA.A$_F$, a low level of GyrA dimer, encoded by the endogenous *gyrA* gene could explain the low level of supercoiling we observe. In the case of our heterodimer preparations, both GyrA (untagged) and GyrBA fusion (his-tagged) are co-expressed in *E. coli*. Various dimeric combinations are therefore obtained in the crude extract before purification. Since the nickel column step pulls down only the GyrBA fusion, GyrA dimers are purified out. It is also possible that some heterodimers contain the His-tagged GyrBA fusion dimerized with a GyrA subunit encoded by the endogenous *gyrA*, even though the expression level of recombinant GyrA is very high (*Figure 1—figure supplement 1a*), which should minimize their formation. Nonetheless, such complexes are expected to occur during expression and to be co-purified. We observed a low level of GyrA cleavage-dependent labeling with our BA.A$_F$ preparation, which demonstrate the existence of these complexes, which we denote BA.A$_{endogenous}$. These complexes can explain the low level of supercoiling observed with the BA$_F$.A$_{59}$ heterodimer, which cannot reconstitute supercoiling activity by subunit exchange, since the contaminating BA$_F$.A$_{endogenous}$ would reconstitute a GyrA$_{endogenous}$ dimer, which is active in the presence of GyrB. It could also underlie the low level of supercoiling observed with the BA.A$_F$ heterodimer in the presence of GyrB, although such activity could also arise from the reconstituted BA fusion dimer. In Gubaev et al. no experiments were done to address whether cleavage originated from the expected recombinant subunit as opposed to possible minor contamination with unmutated, endogenous subunits (*Gubaev et al., 2016*).

Our purification system is not as stringent as in Gubaev et al. We use a single tag on the BA fusion and the expression strategy minimizes the formation of unwanted heterodimers with endogenous GyrA. However, we account for contamination and have established that they are minimal thereby reconstituting a weak supercoiling activity. In Gubaev et al. an extra tag on GyrA is used for purification and should further minimize endogenous contamination (*Gubaev et al., 2016*). However, a small contamination is not excluded. We suggest a cleavage radiolabeling of subunits could have been done to address this. If the recombinant GyrA is mutated for cleavage no labeling should occur and the observation of even a very minor GyrA signal suggests contamination. Our GyrA preparations are usually not contaminated by endogenous GyrA. For instance, our preparation of GyrA$_{I174A}$ does not show any supercoiling activity (nor cleavage) in the presence of GyrB. Our GyrB preparation on the other hand can be contaminated by a low amount of GyrA, undetectable with monoclonal antibodies, as evidenced by the very low level of supercoiling induced by GyrB alone at high concentration (in assays, lower concentrations are used to prevent this unwanted activity). This amount of contaminant GyrA can vary from preparation to preparation. It is therefore possible that the heterodimer preparation is contaminated by a very low amount of endogenous GyrA that escapes the tandem tag purification, the GyrB domain of the fusion interacting with GyrA. In addition, we have demonstrated that GyrB oligomerizes. Therefore, two heterodimers such as BA$_{59}$.A$_F$ and BA$_F$.A$_{endogenous}$ could oligomerize through their GyrB domain (which is unmutated) and the tandem tag purification of BA$_{59}$.A$_F$ could indeed co-purify a small amount of BA$_{59}$.A$_{endogenous}$. In Gubaev et al. the authors trust that the tandem purification procedure will not produce contamination important enough to explain their results without really testing it (*Gubaev et al., 2016*). Their only experiment directly addressing contamination is described in their Supplementary Figure 6 (*Germe et al., 2018*). In this experiment it is shown that *B. subtilis* GyrB can produce active gyrase with an *E. coli* GyrA dimer. However, *E. coli* GyrB cannot reconstitute an active gyrase with *B. subtilis* GyrA dimer (*Gubaev et al., 2016*). They then add either *B. subtilis* or *E. coli* GyrB to a *B. subtilis* BA.A$_F$ heterotetramer and show that only the *B. subtilis* GyrB is capable of reconstituting supercoiling activity. They conclude, rightfully, that their *B. subtilis* heterodimer preparation, expressed in *E. coli*, is not contaminated by *E. coli* GyrA dimers, since these would reconstitute active gyrase with added *E. coli* GyrB. However, this experiment does not exclude the presence of BA.A$_{endogenous}$ where the BA fusion is *B. subtilis* and the GyrA subunit is endogenous *E. coli* from the expression strain. This complex could still undergo

subunit exchange since the BA fusion side is *B subtilis*, producing an *E. coli* GyrA dimer that can reconstitute gyrase activity in the presence of *B. subtilis* GyrB, as shown in their previous experiment. When *E. coli* GyrB is added to the *B. subtilis* $BA.A_{endogenous}$ it cannot promotes subunit exchange as it presumably cannot oligomerize with the *B. subtilis* GyrB domain of the fusion.

## Heterodimers mutated on two sides, such as $BA_F.A_{59}$: Contamination versus partial interface swapping and the role of GyrB flexibility and the free GyrB dimers

Heterodimer designed to have mutations on both sides have been purified by us and others (*Gubaev et al., 2016*). Usually, the catalytic tyrosine is mutated on one side, which abolishes cleavage on this side. The other side bears another mutation, which abolishes supercoiling activity when present on both sides in a homodimer. The CTD is involved in wrapping the DNA in a positive loop prior to strand passage. Therefore, its complete absence from the gyrase complex will abolish supercoiling. However, losing the CTD on one side only of a dimer with two catalytic tyrosine does not abolish supercoiling activity as the dimer can use the single CTD for wrapping, as predicted by the strand-passage model. The supercoiling activity goes down only slightly, by a factor of two. However, if a catalytic tyrosine is mutated on the other side of the missing CTD, a drastic drop of supercoiling activity is observed, suggesting having two catalytic tyrosines is indeed important (see main discussion). These results were obtained in the following configuration: $BA_{59}.A_F$ + free GyrB. Similarly, one can mutate the ATP hydrolysis activity on one side (with the E44Q mutation on the GyrB subunit) and only slightly diminish the supercoiling activity, confirming earlier observations that hydrolysis of only one ATP is necessary for strand passage by type II topoisomerases (*Hartmann et al., 2017*). The authors analyzed the analogous configuration: $BA_Q.A_F$ + free GyrB. Interestingly, the supercoiling activity observed is much less reduced that in the $BA_{59}.A_F$ + free GyrB configuration (we surmise that it would be still lower than the configuration with two catalytic tyrosines, although no quantitative measurements are provided). The swiveling model does not account for this difference, whereas interface swapping (IS) does since we have shown that having two CTDs is important for IS. The $BA_Q.A_F$ has both CTDs and therefore is expected to be more efficient for IS and consistently, shows more efficient reconstitution of supercoiling activity. The authors have also tested the configuration: $BA_Q.BA_F$, where both sides are BA fusion and no free GyrB is added. The supercoiling activity is much reduced, consistent with our observation that constraining GyrB flexibility diminishes supercoiling efficiently. In addition, the absence of free GyrB could also affect IS since we have shown that an excess of GyrB favors IS, potentially through oligomerization of the free GyrB subunits with the heterodimer constructs. Therefore, in the case of $BA_Q.BA_F$, the lack of free GyrB subunits could reduce the efficiency of IS, thereby reducing the efficiency of supercoiling. Therefore, variation of IS amongst heterodimer constructs can underly their difference in supercoiling activity.

However, when these heterodimers undergo *complete* (we emphasize) subunit exchange, meaning all three interfaces are broken and exchange to reconstitute two gyrase dimers separated in solution, the resulting dimers are expected to be both inactive. This was argued to exclude complete subunit exchange as an explanation for the reconstitution of supercoiling by heterodimers with one catalytic tyrosine mutated. However, our experiments suggest that the DNA interface only is exchanged and probably occurs within multimers of the heterodimer, without producing free exchanging gyrase dimers. Therefore, in the case of $BA_Q.A_F$ and $BA_Q.BA_F$, it is possible that only the DNA gate is exchanged, whereas the GyrB subunits are not and retain strand capture activity within the oligomer. The oligomerization of a number of active gyrases could therefore result in a super-complex within which DNA gates can be exchanged, with strand-capture activity provided by a neighboring gyrase. It would be interesting to test our LLL mutant with the E44Q mutation introduced on the other side of the heterodimer. We would expect that the LLL mutation favors DNA gate interface swapping and therefore increase reconstitution of double-strand DNA cleavage. If this double-strand cleavage activity correlates with the reconstitution of supercoiling activity, we would conclude that the exchanged interface can use the ATPase from a neighboring complex for strand capture activity. In addition, interface swapping of a contaminating heterodimer including endogenous GyrA (like $BA_Q.A_{endogenous}$), expected to be efficient, could also account for the significant supercoiling activity observed. In contrast, in the case of the CTD mutant, which is on GyrA and away from the interface, the cleavage activity is reconstituted by exchanged DNA interfaces which are lacking CTDs on both sides. Our results showed that this exchange does not result in increased supercoiling activity. Therefore, the exchanged DNA gates cannot use CTD that

are not directly attached to the exchanged interface. We therefore conclude that, in the case of the CTD mutant, only interface swapping of a contaminating heterodimer including endogenous GyrA ($BA_{59}.A_{endogenous}$ or $BA_F.A_{endogenoous}$) accounts for the observed weak supercoiling activity. Again, the level of supercoiling roughly seems to correlate with the level of expected IS; although this would warrant careful quantitation and analysis.

