## [Editor Report · eLife assessment]

This is an **important** study on DNA gyrase that provides further evidence for its mode of action via a double-stranded DNA break and against a recently-proposed alternative mechanism. The evidence presented is **solid** and is derived from state-of-the-art techniques. The work casts new light on the interactions that occur between gyrase molecules and will be of interest to biochemists and cell biologists.

---

## [Referee Report · Reviewer #1 (Public Review)]

Germe and colleagues have investigated the mode of action of bacterial DNA gyrase, a tetrameric GyrA2GyrB2 complex that catalyses ATP-dependent DNA supercoiling. The accepted mechanism is that the enzyme passes a DNA segment through a reversible double-stranded DNA break formed by two catalytic Tyr residues-one from each GyrA subunit. The present study (now described in a revised manuscript) sought to understand an intriguing earlier observation that gyrase with a single catalytic tyrosine that cleaves a single strand of DNA, nonetheless has DNA supercoiling activity. This unexpected finding led to the proposal that gyrase acts instead via a nicking closing mechanism. Germe et al used bacterial co-expression to make the wild-type and mutant heterodimeric BA(fused).A complexes with only one catalytic tyrosine. Whether the Tyr mutation was on the A side or BA fusion side, both complexes plus GyrB reconstituted fluoroquinolone-stabilised double-stranded DNA cleavage and DNA supercoiling activity. This indicates that the preparations of these complexes sustain double strand DNA passage as envisaged in the current double-strand break mechanism of gyrase. Of possible explanations for how double-strand cleavage arises, contamination of heterodimeric complexes or GyrB with GyrA dimers was ruled unlikely by the meticulous prior analysis of the proteins on native Page gels, by analytical gel filtration and by mass photometry (although low levels of endogenous GyrA were seen in some preparations). Involvement of an alternative nucleophile on the Tyr-mutated protein was ruled out by analysis of mutagenesis studies focused on the catalytic ArgTyrThr triad of residues. Similarly, analysis of 5'- and 3'- DNA ends generated by cleavage ruled out water as a nucleophile. Instead, results of the present study favour a third explanation wherein double-strand DNA breakage arises as a consequence of subunit (or interface/domain) exchange. The authors showed that although the A subunits in the GyrA dimer were thought to be tightly associated, addition of GyrB to heterodimers with one catalytic tyrosine stimulated DNA cleavage with a time lag consistent with rapid DNA-dependent subunit or interface exchange to generate complexes with two catalytic tyrosines capable of double-stranded DNA breakage. Subunit exchange between heterodimeric complexes was facilitated by DNA bending and wrapping by gyrase, by the ability of both GyrA and GyrB to form higher order aggregates and by dense packing of gyrase complexes on DNA. By addressing a puzzling paradox, this study provides further support for the accepted double strand break (strand passage) mechanism of gyrase (without having to invoke a nicking-closing mechanism) and opens new insights on subunit exchange that may have biological significance in promoting DNA recombination and genome evolution.

The conclusions of the work are mostly well supported by the experimental data. Moreover, in the revised manuscript, the various concepts, experiments and outcomes are better explained and more accessible to the reader through a reorganised text, clearer figures and an extended Supplementary section.

Strengths:

The study examines a fundamental biological question, namely the mechanism of DNA gyrase, an essential and ubiquitous enzyme in bacteria, and the target of fluoroquinolone antimicrobial agents.

The experiments have been carefully done and the analysis of their outcomes is comprehensive, thoughtful and considered.

The work uses an array of complementary techniques to characterize preparations of GyrA, GyrB and various gyrase complexes. In this regard, mass photometry seems particularly useful. Analysis revealed that purified GyrA and GyrB can each form multimeric complexes and highlights the complexities involved in investigating the gyrase system.

The various possible explanations for the double-strand DNA breakage by gyrase heterodimers with a single catalytic tyrosine are considered and addressed by appropriate experiments.

The study highlights the potential biological importance of interactions between gyrase complexes through domain-or subunit-exchange.

---

## [Referee Report · Reviewer #2 (Public Review)]

DNA gyrase is an essential enzyme in bacteria that regulates DNA topology and has the unique property to introduce negative supercoils into DNA. This enzyme contains 2 subunits GyrA and GyrB, which forms an A2B2 heterotetramer that associates with DNA and hydrolyzes ATP. The molecular structure of the A2B2 assembly is composed of 3 dimeric interfaces, called gates, which allow the cleavage and transport of DNA double stranded molecules through the gates, in order to perform DNA topology simplification.

The article by Germe et al. questions the existence and possible mechanism for subunit exchange in the bacterial DNA gyrase complex.

The complexes are purified as a dimer of GyrA and a fusion of GyrB and GyrA (GyrBA), encoded by different plasmids, to allow the introduction of targeted mutations on one side only of the complex. The conclusion drawn by the authors is that subunit exchange does happen in vitro, favored by DNA binding and wrapping. They propose that the accumulation of gyrase in higher-order oligomers can favor rapid subunit exchange between two active gyrase complexes brought into proximity. This study is nicely illustrated with diagrams that explain the possible mechanism.

The authors are also debating the conclusions of a previous article by Gubaev, Weidlich et al 2016 (https://doi.org/10.1093/nar/gkw740). Gubaev et al. originally used this strategy of complex reconstitution to propose a nicking-closing mechanism for the introduction of negative supercoils by DNA gyrase, an alternative mechanism that precludes DNA strand passage, previously established in the field. Germe et al. propose that the detected negative supercoiling activity in this earlier study may be due to the subunit swapping of the recombinant protein with the endogenous enzyme.

Strengths

The mix of gyrase subunits is plausible, this mechanism has been suggested by Ideka et al, 2004 and also for the human Top2 isoforms with the formation of Top2a/Top2b hybrids being identified in HeLa cells (doi: 10.1073/pnas.93.16.8288).

Germe et al have used extensive and solid biochemical experiments, together with thorough experimental controls, involving :

- the purification of gyrase subunits including mutants with domain deletion, subunit fusion or point mutations.

- DNA relaxation, cleavage and supercoiling assays

- biophysical characterization in solution (size exclusion chromatography, mass photometry, mass spectrometry)

Together the combination of experimental approaches provides convincing evidence for subunit swapping in gyrase in vitro, despite the technical limitations of standard biochemistry applied to such a complex macromolecule.

Weaknesses

The conclusions of this study could be strengthened by in vivo data to identify subunit swapping in the bacteria. Indeed, if shown in vivo, together with this biochemical evidence, this mechanism could have a substantial impact on our understanding of bacterial physiology and resistance to drugs. These in vivo perspectives are beyond the scope of the present in vitro investigation but are however explained by the authors.

---

## [Author Response]

Note to the editor and reviewers.

All the authors would like to thank the editorial team and the two anonymous reviewers for their efforts and thoughtfulness in assessing our manuscript. We very much appreciate it and we all believe that the manuscript has been much improved in addressing the comments and suggestions made.

General considerations on the revised manuscript

We have applied extensive modifications to the manuscript with our main goal being the improvement of clarity. The Introduction has been changed mainly to introduce precisely our terminology and we have stuck to it in the rest of the manuscript. The Results section has been divided up into more defined sections. The discussion has been extensively re-written to improve clarity, following the suggestion of the reviewers. Main figures 1 and 4 have been modified with clearer schematics. Supplementary figures and legends have been modified and several supplementary schematic figures have been added to clearly present our interpretations for various data. We have added a Supplementary Discussion where the most detailed technical parts of our discussion are presented to avoid unnecessarily weighing down the main discussion, where our main conclusions are outlined. We have presented our mass photometry mixing experiment in a new supplementary figure, with detailed explanation. We have also expanded our discussion of in vivo and general relevance of our study.

Response to manuscript evaluation

Our manuscript has been evaluated as a valuable study and presenting solid experimental evidence. We appreciate the recognition of our work.

Two weaknesses were identified by reviewers: (1) our experiments do not completely exclude the possibility of an alternative nucleophile. This relates to the evaluation of our experimental evidence. (2) Our study does not address the in vivo relevance of the interface swapping phenomenon, which relate to the value of the study for the community.

Response to the evaluation of experimental evidence (Weakness #1):

We argued in the original manuscript that we have excluded completely the presence of an alternative nucleophile. This conclusion is based on a series of experiments which were presented in the originally submitted manuscript. These experiments are not discussed by the reviewers in relation to this main conclusion and therefore we suggest that they have not been properly evaluated. We believe our conclusion to be appropriately supported by these data (see our response to reviewer #1). In addition, the criticism of our gel-filtration data by reviewer #2 was based on a misinterpretation of Supplementary figure 1 b. We accept of course that the way the data was presented could be misleading and we assume responsibility for this. We have attempted to correct this by changing the main text and the figures legends and annotation. In conclusion, we believe that the evaluation of experimental evidence as presented in the revised manuscript could be upgraded to “convincing”.

Response to our study general relevance evaluation (weakness #2):

We agree with both reviewers about the in vivo relevance of our observation being an important question, not addressed so far. Indeed, the value of our study would be greatly increased by in vivo data and be of interest to a wider audience. However, we would like to argue that our study would interest a wider audience than initially stated for the following reasons: (1) Our study is the first evidence of interface swapping in vitro and will constitute a base to investigate this phenomenon both in vivo and in vitro. It will therefore interest a wide audience due to the potential involvement of interface swapping in a wide range of processes, such as recombination, evolution, and drug targeting (see also below). (2) DNA cleavage is the central mode of action of antibiotics targeting bacterial type II topoisomerases (i.e. topoisomerases “poisons”). This already established target is one of the few having produced new scaffolds and too few new antibacterial are in production to fulfill medical needs. The role of interface stability is also emerging as a modulator of the efficiency of topoisomerase poisons. See for instance (Germe, Voros et al. 2018, Bandak, Blower et al. 2023). By shedding light on interface dynamics, our study will be of interest to scientist interested in the development of these drugs. In addition, the heterodimer system can potentially produce detailed mechanistic information (Gubaev, Weidlich et al. 2016, Hartmann, Gubaev et al. 2017, Stelljes, Weidlich et al. 2018) not only on gyrase but also on other, dimeric type II topoisomerases or even other dimeric enzyme in general. We have amended the manuscript to make these points clearer. Therefore, we believe that the evaluation of the revised manuscript’s relevance could be upgraded to “important”.

Point-by-point response to the reviewer

**Reviewer #1 (Public Review):**
Germe and colleagues have investigated the mode of action of bacterial DNA gyrase, a tetrameric GyrA2GyrB2 complex that catalyses ATP-dependent DNA supercoiling. The accepted mechanism is that the enzyme passes a DNA segment through a reversible double-stranded DNA break formed by two catalytic Tyr residues-one from each GyrA subunit. The present study sought to understand an intriguing earlier observation that gyrase with a single catalytic tyrosine that cleaves a single strand of DNA, nonetheless has DNA supercoiling activity, a finding that led to the suggestion that gyrase acts via a nicking closing mechanism. Germe et al used bacterial co-expression to make the wild-type and mutant heterodimeric BA(fused). A complexes with only one catalytic tyrosine. Whether the Tyr mutation was on the A side or BA fusion side, both complexes plus GyrB reconstituted fluoroquinolone-stabilized double-stranded DNA cleavage and DNA supercoiling. This indicates that the preparations of these complexes sustain double strand DNA passage. Of possible explanations, contamination of heterodimeric complexes or GyrB with GyrA dimers was ruled out by the meticulous prior analysis of the proteins on native Page gels, by analytical gel filtration and by mass photometry. Involvement of an alternative nucleophile on the Tyr-mutated protein was ruled unlikely by mutagenesis studies focused on the catalytic ArgTyrThr triad of residues. Instead, results of the present study favour a third explanation wherein double-strand DNA breakage arises as a consequence of subunit (or interface/domain) exchange. The authors showed that although subunits in the GyrA dimer were thought to be tightly associated, addition of GyrB to heterodimers with one catalytic tyrosine stimulates rapid DNA-dependent subunit or interface exchange to generate complexes with two catalytic tyrosines capable of double-stranded DNA breakage. Subunit exchange between complexes is facilitated by DNA bending and wrapping by gyrase, by the ability of both GyrA and GyrB to form higher order aggregates and by dense packing of gyrase complexes on DNA. By addressing a puzzling paradox, this study provides support for the accepted double strand break (strand passage) mechanism of gyrase and opens new insights on subunit exchange that may have biological significance in promoting DNA recombination and genome evolution.The conclusions of the work are mostly well supported by the experimental data.Strengths:The study examines a fundamental biological question, namely the mechanism of DNA gyrase, an essential and ubiquitous enzyme in bacteria, and the target of fluoroquinolone antimicrobial agents.The experiments have been carefully done and the analysis of their outcomes is comprehensive, thoughtful and considered.The work uses an array of complementary techniques to characterize preparations of GyrA, GyrB and various gyrase complexes. In this regard, mass photometry seems particularly useful. Analysis reveals that purified GyrA and GyrB can each form multimeric complexes and highlights the complexities involved in investigating the gyrase system.The various possible explanations for the double-strand DNA breakage by gyrase heterodimers with a single catalytic tyrosine are considered and addressed by appropriate experiments.The study highlights the potential biological importance of interactions between gyrase complexes through domain-or subunit-exchange

We thank the reviewer for their support, effort, and comments. The above is a great summary.

Weaknesses:The mutagenesis experiments described do not fully eliminate the perhaps unlikely participation of an alternative nucleophile.

We agree that the mutagenesis experiment on its own does not fully eliminate the possibility of an alternative nucleophile. The number of residues mutated is limited, and therefore it is possible we have missed a putative alternative nucleophile.

However, we have other data and experiments supporting the conclusion that no alternative nucleophile exists. Therefore, we want to stress that our conclusion that no such alternative exist is based on these extra data. These data and experiments are not discussed by either reviewer despite being present in the original manuscript. This puzzled us and we have modified the manuscript and the figures in the hope that they, and their significance, would not be missed.

Briefly:

1. We have performed cleavage-based labeling of the nucleophile responsible for cleavage. This experiment is depicted in Figure 4. The nucleophilic activity of the residue involved results in covalent link between the polypeptide (that includes the residue) and radiolabeled DNA. Therefore, a polypeptide that includes an active nucleophile will be radiolabeled and visible, whereas a polypeptide that is missing an active nucleophile will remain unlabeled and invisible. We can distinguish the BA and the A polypeptide from their size. In the case of the BA.A complex both the BA polypetide and the A polypetide are radiolabeled and therefore both have an active nucleophile. In the case of the BAF.A complex, the unmutated A polypeptide is labeled, meaning that a nucleophile is still active. In contrast, the BAF polypeptide shows no detectable labeling. This result means that removing the hydroxyl group from the catalytic tyrosine abolishes any protein-DNA covalent link, suggesting that no other nucleophile from the BA polypetidic chain can substitute for the catalytic tyrosine hydroxyl group. This experiment excludes the possibility of an alternative nucleophile coming from the polypeptidic chain of either GyrA or GyrB. This experiment, described in figure 4, is not discussed by the reviewer. This experiment is similar in principle to early experiments identifying catalytic tyrosine in topoisomerases. See for instance, (Shuman, Kane et al. 1989).

2. The experiment above does not exclude a nucleophile coming from the solvent. To exclude this possibility, we have used T5 exonuclease (which needs a free 5’ DNA end to digest) and ExoIII (which need a free 3’ DNA end to digest). We have shown the reconstituted cleavage is not sensitive to T5 and sensitive to ExoIII. This shows that the 5’ end of the cleaved sites are protected by a bulky polypeptide impairing T5 activity, which is active in our reaction as shown by the digestion of a control DNA fragment. This experiment shows that the reconstituted cleavage is very unlikely to come from a small nucleotide potentially provided by the solvent. This experiment is described in the main text and the results are shown in supplementary figure 5. It is not mentioned by either reviewer.

3. Finally, we would like to emphasize our experiment comparing the BAF.A59 to BALLL.A59. The BALLL.A59 complex displays increased cleavage compared to BAF.A59. If this increased cleavage was due to an alternative nucleophile on the BALLL side, we would expect an accompanying increase in supercoiling activity since the BALLL.A59 possesses one CTD, which is sufficient for supercoiling. The fact that no increased supercoiling activity is observed strongly suggests subunit exchange reconstituting an A59 dimer, inactive for supercoiling but active for cleavage. We believe this somewhat complex observation to be quite significant and we have attempted to clarify the manuscript and discuss its full significance in several places.

**Reviewer #1 (Recommendations For The Authors):**
An interesting paper on DNA gyrase that explains a puzzling paradox in terms of the double-strand break mechanism.Major points1. The authors consider several mechanisms that could potentially explain their data. On page 15, the authors present the evidence against the nicking closing mechanism proposed by Gubaev et al. Throughout the manuscript, they indicate where their experimental results agree with this earlier work but should also indicate and account for differences. For example, Gubaev et al describe cross linking experiments that they claim rule out subunit exchange. These aspects should be clearly explained.

Thank you for the suggestion. We have re-written the discussion to address this point. We are extensively discussing experiments from (Gubaev, Weidlich et al. 2016), and offer our interpretation of apparently conflicting results. We suggest that their experiments are basically consistent with our data when correctly interpreted. To keep the main manuscript clear, we have added a supplementary discussion where experiments from (Gubaev, Weidlich et al. 2016) are discussed further in relation to our data.

1. Page 9. The experiments done to rule out the perhaps unlikely alternative nucleophile hypothesis relate to the possible role of the Arg and Threonine of the RYT triad. These residues are close to the DNA and therefore are prime candidates and attractive targets for mutagenesis. However, strictly speaking, the mutant enzyme data presented do not rule all possibilities. For example, Serine is often the nucleophile used by resolvases to effect DNA recombination via subunit exchange. The ideal experiment to rule out/rule in other nucleophiles would be to identify the residue(s) that become attached to DNA in the cleavage reaction.

Please see above. We have effectively ruled an alternative nucleophile with our cleavage-based labeling experiment and others that were present and discussed in the original manuscript but were missed. We have modified the manuscript and figures in order to make this point clearer than before.

1. p17. The readout for subunit exchange used by the authors is double-stranded DNA cleavage. Attempts to directly detect the formation of the DNA cleaving complexes GyrA2B2 and (GyrBA)2 (arising from subunit exchange between heterodimers) by mass photometry were not successful. Perhaps FRET would have been another approach to try as it could also detect interface and domain interchanges.

Directly detecting interface exchange directly by proximity experiment would be extremely useful. FRET would have to be done in the BAF.A + GyrB configuration where the amount of interface exchange is important. Now, we do not have the tools to do that and developing them would be outside the scope of the study. We propose cross linking experiment to be done in the future. We argue that the manuscript is convincing without these for now. This will be addressed in the future. This point, and other possible future experiments are now discussed in the discussion section.

1. The underlying canvas of this paper is the strand passage mechanism of gyrase. It would seem appropriate to include the papers first proposing it - Brown P.O and Cozzarelli N.R. (1979) and Mizuuchi K et al (1980).

We very much agree. These papers have now been added in the introduction as appropriate, highlighting the relationship between double-strand cleavage and the strand-passage mechanism.

1. Figure 1. The quality of the insets is poor. It is difficult to pick out the key catalytic residues and their disposition vis-a-vis DNA.

We agree, Figure 1 has been re-done and the schematic theme has been harmonized throughout the whole manuscript. We very much hope that clarity has improved. Thank you for the suggestion.

1. The experimental work is a very detailed analysis of a specific feature of engineered gyrase heterodimers. Making the work accessible to the general reader will be important. Using shorter paragraphs each with a specific theme might help. In particular, the second paragraph of the Results on p7, the section on p9 and bottom of p11, p13 and the first paragraph of the Discussion on p14 are each a page or more long. A shorter manuscript that avoids overinterpretation of the smaller details would also help.

We agree. We have now split long paragraphs into individual sections, with titles, in the Results. This structure is recapitulated at the beginning of the discussion, and we have split the discussion into shorter paragraphs, each with a unique point being made.

1. The impact of the Gubaev et al (2016) paper for the field in general, and as the catalyst for the present work should be better documented. Mention of this earlier paper and its significance at the beginning of the Abstract and elsewhere e.g in the Introduction might also help with a more logical organization of the current findings and result in a shorter paper (which would be easier to read).

We have added a reference to (Gubaev, Weidlich et al. 2016) in the abstract and have expanded our introduction

Minor points1. Legends for Figs 2 and 6; Supplementary Figs 1 and 8. The designation of subfigures as a, b, c, d , e etc appears to be incorrect. Check throughout and in the text.

The manuscript has been checked for such errors.

1. Figure 2, and first paragraph p8. Peaks in Fig 2c should be labelled to facilitate discussion on p8.

Agreed, this has been done.

1. Supplementary Fig 4 and elsewhere in the manuscript. A variety of notations are used to denote phenylalanine mutants e.g. AsubscriptF, AsuperscriptF and AF. Check and use one format throughout.

Done

1. Figures showing gels include the label '+EtBr, +cipro'. This is somewhat confusing because EtBr was contained in the gel (not the samples) whereas cipro was included in the reaction. Modify or describe in the legend..

We have re-written the figure legend.

1. Supplementary Fig 4b describes a small effect on the ratio of linear to nicked DNA for the triple LLL mutant. Is this significant? How many times was the measurement made?

This has been addressed in the original manuscript in the supplementary data. In term of quantification, the experiment has been done 3 times for each prep, with the same GyrB prep and concentration. The standard error is displayed on the figure. This result is very reproducible and have been reproduced more than 3 times. No LLL cleavage assay showed more single-strand than double-strand cleavage. For the phenylalanine mutant, no cleavage assay showed more double-strand than single-strand cleavage.

1. Supplementary Fig 5 legend. Should 'L' read 'size markers' (and give their sizes)?

Yes indeed, we have modified the figure to clarify.

1. p11 line 5. Is this statement correct?

Yes, it is correct. Although we hope we are on the same line. When the Tyrosine is mutated on one side only of the heterodimer, both single- and double-strand cleavage are protected from T5 exonuclease digestion.

1. 12 last line should read...and supercoiling activity (not shown)..were

Thank you, done.

There are a number of typos throughout the text, for example:

Page 3 line..Difficult to conclude...what?

Page 3 para 3...Lopez....and Blazquez

We have corrected these typos and checked the whole manuscript.

**Reviewer #2 (Public Review):**
DNA gyrase is an essential enzyme in bacteria that regulates DNA topology and has the unique property to introduce negative supercoils into DNA. This enzyme contains 2 subunits GyrA and GyrB, which forms an A2B2 heterotetramer that associates with DNA and hydrolyzes ATP. The molecular structure of the A2B2 assembly is composed of 3 dimeric interfaces, called gates, which allow the cleavage and transport of DNA double stranded molecules through the gates, in order to perform DNA topology simplification.The article by Germe et al. questions the existence and possible mechanism for subunit exchange in the bacterial DNA gyrase complex.The complexes are purified as a dimer of GyrA and a fusion of GyrB and GyrA (GyrBA), encoded by different plasmids, to allow the introduction of targeted mutations on one side only of the complex. The conclusion drawn by the authors is that subunit exchange does happen, favored by DNA binding and wrapping. They propose that the accumulation of gyrase in higher-order oligomers can favor rapid subunit exchange between two active gyrase complexes brought into proximity.The authors are also debating the conclusions of a previous article by Gubaev, Weidlich et al 2016 (https://doi.org/10.1093/nar/gkw740). Gubaev et al. originally used this strategy of complex reconstitution to propose a nicking-closing mechanism for the introduction of negative supercoils by DNA gyrase, an alternative mechanism that precludes DNA strand passage, previously established in the field. Germe et al. incriminate in this earlier study the potential subunit swapping of the recombinant protein with the endogenous enzyme, that would be responsible for the detected negative supercoiling activity.Accordingly, the authors also conclude that they cannot completely exclude the presence of endogenous subunits in their samples as well.StrengthsThe mix of gyrase subunits is plausible, this mechanism has been suggested by Ideka et al, 2004 and also for the human Top2 isoforms with the formation of Top2a/Top2b hybrids being identified in HeLa cells (doi: 10.1073/pnas.93.16.8288).Germe et al have used extensive and solid biochemical experiments, together with thorough experimental controls, involving :the purification of gyrase subunits including mutants with domain deletion, subunit fusion or point mutations.DNA relaxation, cleavage and supercoiling assaysbiophysical characterization in solution (size exclusion chromatography, mass photometry, mass spectrometry)Together the combination of experimental approaches provides solid evidence for subunit swapping in gyrase in vitro, despite the technical limitations of standard biochemistry applied to such a complex macromolecule.

We thank the reviewer for their supportive and considered comments.

WeaknessesThe conclusions of this study could be strengthened by in vivo data to identify subunit swapping in the bacteria, as proposed by Ideka et al, 2004. Indeed, if shown in vivo, together with this biochemical evidence, this mechanism could have a substantial impact on our understanding of bacterial physiology and resistance to drugs.

Thank you for this comment. Indeed, whether this interface exchange can happen in vivo and lead to recombination is a very important question. However, we believe that this is outside the scope of this study simply because of the amount of work one can fit into one paper. Proving that interface exchange can happen in vitro has already necessitated a number of non-trivial experiments and likewise investigating interface exchange in vivo will require a careful, long-term study (see our reply to reviewer #2 comment, who also raised this point). We can’t address it with one additional experiment with the tools we have. However, we very much hope to do it in the future.

**Reviewer #2 (Recommendations For The Authors):**
Specific questions and comments for the authors:1. Complex identification during purificationThe statement line 236-237 that "Our heterodimer preparation showed a single-peak on a gel-filtration column, distinct from the GyrA dimer peak" is not entirely clear. In Fig supp 1 b, how can the authors conclude from the superose 6 that GyrBA is separated from the GyrA dimer? Since they seem close in size 160/180kDa, they are unlikely to be well separated in a superose 6 gel filtration column. The SDS-PAGE seems to show both species in the same fractions #15-17 therefore it would not be possible to distinguish GyrBA. A from A2.

There appears to be some confusion about what Supp Fig. 1b shows. First, in all our gel filtration conditions both GyrBA and GyrA can’t exist as monomers at a significant concentration. Therefore, we can never observe the GyrBA monomer on a gel filtration column. Supp Fig. 1b shows the gel filtration profile of the BA.A heterodimer only. This is the output of the last, polishing step in the reaction. We analyze these results using SDS-PAGE. Therefore, the BA.A heterodimer will be denatured and separated into 2 polypeptides: GyrBA and GyrA, which migrates according to their size in an SDS-PAGE and forms two bands. These two bands do not represent two separate species in solution. They represent the separation of one species only, the BA.A heterodimer into its two, denatured, subunits: GyrA and GyrBA. We do not conclude from Supp Fig. 1 as a whole that GyrBA and the GyrA dimer are well separated, and this is not stated in the manuscript. We conclude that the BA.A dimer is fairly well separated from the GyrA dimer. They have significant different size (~260 kDa and ~180 kDa respectively) and form different peaks on a gel filtration column. The BA.A heterodimer has a GyrA subunit and therefore will shows a GyrA band on an SDS-PAGE, like the GyrA dimers but the two are obviously distinct in their quaternary structure. We are hoping that our new schematics and re-write of some of the results and figure legends will clarify this.

Panel 6 shows a different elution volume for the 2 species BA.A and A2 on an analytical S200 column, which appears better at separating the complexes in this size range.Did the authors consider using a S200 column instead of superose 6 for the sample preparation, to optimize the separation of GyrBA. A from A2?

This is not a necessarily true statement (see above). We have not run the GyrA dimer on a Superose 6 column. The analysis was done on an s200 because extensive data for the GyrA dimer was already available with this, already calibrated column. We do not expect the Superose 6 to be worse in this size range. In fact, it might even be better. The Superose 6 profile in Supp. Fig. 1b shows BA.A only and no GyrA dimer. We have clarified the annotations in the figure to make this clearer.

Regarding the analytical gel filtration experiment, there is however an overlap in the elution volume in the analytical column, therefore how can the authors ensure there is no excess free A2 complex in the GyrBA. A sample?

Indeed, there is an overlap, but we argue that it is overstated. The important part of the overlap is where the maximum height of the GyrA peak is positioned compared to the BA.A trace, not where the traces intersect. This overlap is minimal. If a contaminating GyrA peak was hidden in the BA.A peak, it would have to be at least 10 times less intense than the BA.A peak. Since BA.A and GyrA dimer have roughly the same extinction coefficient, this means that a contamination would detectable at 10 % or even less. Our mass photometry further excludes such contamination.

Alternatively, the addition of a larger (cleavable) tag at the C-terminal end of the BA construct (therefore not disturbing dimer association) could allow to better distinguish the 2 populations already at the size exclusion step.

This is true and could allow cleaner purification. There are also other ways to achieve cleaner purification, like adding a secondary tag. However, like we argue in the manuscript, our contaminations are already minimal. It is questionable what benefits could be gained in changing the protocol. We also argue that the tandem tag method does not completely exclude contamination (Supplementary Discussion) and therefore we are not sure if this would be worth the time and expenditure.

1. GyrA and GyrB Oligomers:In the mass photometry experiment, the authors explain that the low concentration of the proteins promotes dissociation of GyrA dimers, hence the detection of GyrA monomers instead of GyrA dimers, which are also detected in the GyrBA.A sample.However, it cannot be concluded that the GyrA dimer is not formed in the condition of the gel filtration chromatography, at higher concentration.

In our mass photometry experiment, The BA.A sample is not as diluted as the GyrA dimer and much closer to our experimental condition. Since we have calculated the dissociation constant, we can calculate the expected level of dissociation (or reassociation). The level of dissociation is minimal in these conditions. If some dissociation is expected from the BA.A heterodimers, a very low amount of GyrBA monomer should also be present and yet they are not observed. We presume that it is because mass photometry is much more sensitive to GyrA (see our mixing mass photometry experiment that we have added). If the GyrA would reassociate at higher concentration, it would do so either with itself (forming a GyrA dimer) or with the GyrBA monomer, reforming the heterodimer. Assuming both GyrA dimer and heterodimer have the same dissociation constant, roughly one third of the GyrA monomer would reassociate with themselves. Assuming even complete reassociation of the GyrA dimer, this would leave only GyrA dimer accounting for 2% of the prep.

Another interpretation would be to assume that GyrBA monomers are not present at all and that GyrA monomer are reassociating only with themselves. This is not valid because of the following thermodynamic reason:

Since the profile for the GyrA dimer are collected at equilibrium, we should expect a ratio between GyrA monomer and dimers that follow the dissociation constant. In other words, if the GyrA monomer were in equilibrium with GyrA dimer we should expect a much higher dimer concentration already as the GyrA monomers are not as dilute. We do not observe a GyrA dimer peak in the BA.A profile, even though we can detect a low amount of GyrA dimer mixed with BA.A. Therefore, we conclude that the observed GyrA monomer must be in equilibrium with another dimerization partner, which is most probably the GyrBA monomer (see above). Therefore, only a minimal amount of GyrA dimer is expected to be formed at higher concentration by direct reassociation. This could probably increase if we let this solution-based exchange carry on for a long time at dissociation equilibrium. We have actually shown that this solution-based exchange is very slow and take several days because of the low dissociation at equilibrium.

The mass spectrometry analysis in Fig 2 confirms the presence of (monomeric) GyrA in the sample, despite different experimental conditions.

The concentration of heterodimer in the mass spectrometry experiment is actually higher than in the mass photometry experiment. This shows that self-reassociation of the GyrA monomer as suggested above is undetectable with mass spectrometry at higher concentration.

We considered that the “GyrA monomer” peak could be a contaminating GyrB monomer, which is ~90 kDa, which would explain the lack of reassociation. However, the mass spectrometry peak shows precisely the expected molecular weight of GyrA so we interpret this peak as arising from very limited dissociation of the BA.A heterodimer. The reassociation is limited at high concentration due simply to the fact that the difference in concentration between the mass photometry and our other experimental conditions is not that high. The GyrA dimer had to be diluted 400 times to see significant dissociation and yet even at this very low concentration the dissociation is far from complete.

Our general conclusions on the couple of point above is that we cannot completely exclude the presence of GyrA dimers being present, although they are undetectable in our working conditions either by mass photometry (lower concentration), Mass spectrometry (higher concentration) and even gel filtration (even higher concentration, see above). For the mass photometry, we have established that our detection threshold for a contamination is very low (see our mixing experiment).

Figure 2A: the authors state in the introduction that GyrB is a monomer in solution and then explain that the upper bands in the native gel are multimer of GyrB. Could the authors comment and provide the size exclusion profile of the Gyr B purification?

We have expanded our discussion of this. However, we have not been successful in collecting a gel filtration profile for GyrB. This is likely due to excessive oligomerization at the concentration we are using for gel filtration. We suggest that our mass photometry and Blue-Native PAGE experiment shows clearly that GyrB can be detected as a monomer in solution at the appropriate dilution. However, GyrB tends to oligomerize in a regular fashion (Consider especially Supp Fig. 8a), which suggest that it could align heterodimers on DNA in a linear, regular orientation. We have added a discussion of this.

Together the relevance of the oligomeric state of purified GyrA or GyrB should be clarified, relative to their role in subunit swapping.

We have added explanation in our discussion, while also trying to not be too speculative. Basically, we believe that GyrB oligomerization is likely to be involved. It is difficult to conclude for GyrA since no experiment has allowed us to test it. Therefore, the role of GyrA oligomerization, if any, is unclear. The GyrA tetramer is very prominent though and forms very easily. GyrB on the contrary forms longer oligomers more readily than GyrA and we surmise that this would help interface exchange. However, the structure of these GyrA and GyrB oligomers is not clear, which make it difficult to go beyond speculation on this. It would be a very interesting experiment if we were able to suppress GyrB oligomerization whilst conserving its ability to promote strand-passage and cleavage. Same goes for GyrA. Unfortunately, we are unable to do that at this time.

1. Subunit exchangeLine 320: the concept of subunit exchange in this context should be clearly explained. If one understands correctly, the authors mean that the BAF polypeptide, part of the BAF.A complex, could be replaced by a combination of B+A therefore forming a fully functional WT A2B2 gyrase complex.

Thank you for the suggestion. We have harmonized and clearly defined our terminology for interface swapping and subunit exchange in the introduction and attempted to be much more rigorous when referring to it.

A great effort has been done in this study to explain all the pros and cons of the experimental design but the length of the explanations may prevent readers outside of the field to fully appreciate the conclusions. This article would benefit from the addition of a few schematics to summarize the working hypothesis.

Thanks for the suggestion. We have added a series of schematics to illustrate our interpretation for each construct. As mentioned above the terminology has been more rigorously defined and updated throughout the manuscript.

1. Presence of endogenous GyrALine 419-425: it is quite difficult to follow the explanations regarding the possible contamination of the sample by endogenous GyrA.Maybe these points should rather be addressed in the discussion, when debating the conclusions of Gubaev et al.

We agree. We have re-organized the Discussion doing just that. We added a Supplementary Discussion in which we further discuss the contamination problem in relation to (Gubaev, Weidlich et al. 2016).

Production of the subunits in another (non bacterial) expression system or a cell free system may prevent the association of endogenous protein.

Absolutely. We are planning on addressing this in the future, using the yeast expression system.

1. Mechanism for subunit swappingLines 588-595: As described by the authors the BA fusion shows decreased activity when compared with the WT probably due to limited conformational flexibility in absence of an additional linker sequence between the fused subunits.The affinity of BA for A may possibly be reduced compared to the free A2B2 complex, due to a relative stiffness of the fusion upon full association with a free B subunit, as rightfully pointed by the authors.If subunit exchange do happen in vitro, at least in the conditions of this study, the authors could assess the affinity of BA for A, when compared to the association of free B and A subunitsExperiments using analytical ultracentrifugation or surface plasmon resonance (SPR) may allow to determine the relative affinity of the BA +(A+B) compared to the A2B2 complex. This could be done also for the BALLL mutant and association with A59.

It would be extremely useful to measure the affinity of BA for A. However, this is difficult because of the high affinity of the interface. To measure a dissociation constant, one has to be able to measure the concentration of the monomer and the dimer at equilibrium. Because of this, the complex must be diluted enough to see any dissociation, making detection difficult. In practice, this also means that we cannot purify monomeric versions of these subunits. We therefore can’t perform “on-rate” study on an SPR surface, which would require flowing monomers on its partner subunit tethered to the SPR surface. However, we could perform “off-rate” studies, but the dissociation time is likely to be very long, making the measurement difficult. We have not tried it though, and it could turn out to be informative. An analysis of antibodies off-rate done in the past could provide a guideline for us to perform this experiment. Analytical ultracentrifugation is an excellent technique and could in theory provide information. In practice however it would be still necessary to dilute the complex enough to obtain significant dissociation at equilibrium, making detection difficult. As far as we are aware, analytical ultracentrifugation rely on UV absorbance for protein detection and therefore we probably would not detect our material at the necessary dilution. We are however open-minded about technique with very sensitive detection methods that could be used.

1. In vivo relevanceThe study does not conclude on the subunits exchange in vivo, which have been suggested by earlier studies by Ikeda et al. To elaborate further on the relevance of such mechanism in the bacteria, experiments involving the fluorescent labeling of endogenous / exogenous mutant subunits may be required to provide further information on this phenomenon.

We completely agree that the in vivo relevance of such phenomena is the central question. Addressing this directly is not trivial though. Expressing both BA and A in vivo will results in random partnering and lead to a mix of dimers: A2 (1/4), BA2(1/4) and BA.A (1/2), assuming equal interface affinity. Therefore, to see subunit exchange in the same way as in vitro, one would have to get rid of the BA2 and A2 dimer together, or the BA.A dimer only. Our initial strategy to do that would be to engineer a specific dimer as being uniquely targeted for degradation. This could allow us to “get rid” of for instance the BA.A dimer. Subsequently, we would turn off the degradation and translation together and observe the rate of subunit exchange. This is not trivial though and would be the subject of a further study.

1. Figure 3: I guess the "intact" label refers to the supercoiled DNA (SC) ? It also appears as "uncleaved" in supp Figure 6. The same label for this topoisomer should be used throughout.

Thank you for pointing that out. It has now been corrected.

Bandak, A. F., T. R. Blower, K. C. Nitiss, R. Gupta, A. Y. Lau, R. Guha, J. L. Nitiss and J. M. Berger (2023). "Naturally mutagenic sequence diversity in a human type II topoisomerase." Proceedings of the National Academy of Sciences 120(28).

Germe, T., J. Voros, F. Jeannot, T. Taillier, R. A. Stavenger, E. Bacque, A. Maxwell and B. D. Bax (2018). "A new class of antibacterials, the imidazopyrazinones, reveal structural transitions involved in DNA gyrase poisoning and mechanisms of resistance." Nucleic Acids Res.

Gubaev, A., D. Weidlich and D. Klostermeier (2016). "DNA gyrase with a single catalytic tyrosine can catalyze DNA supercoiling by a nicking-closing mechanism." Nucleic Acids Res 44(21): 10354-10366.

Hartmann, S., A. Gubaev and D. Klostermeier (2017). "Binding and Hydrolysis of a Single ATP Is Sufficient for N-Gate Closure and DNA Supercoiling by Gyrase." J Mol Biol 429(23): 3717-3729.Shuman, S., E. M. Kane and S. G. Morham (1989). "Mapping the active-site tyrosine of vaccinia virus DNA topoisomerase I." Proc Natl Acad Sci U S A 86(24): 9793-9797.

Stelljes, J. T., D. Weidlich, A. Gubaev and D. Klostermeier (2018). "Gyrase containing a single C-terminal domain catalyzes negative supercoiling of DNA by decreasing the linking number in steps of two." Nucleic Acids Res.